# No Free Lunch from Deep Learning in Neuroscience: A Case Study through Models of the Entorhinal-Hippocampal Circuit

**Rylan Schaeffer**
Computer Science
Stanford
rschaef@cs.stanford.edu

**Mikail Khona**
Physics
MIT
mikail@mit.edu

**Ila Rani Fiete**
Brain and Cognitive Sciences
MIT
fiete@mit.edu

## Abstract

Research in Neuroscience, as in many scientific disciplines, is undergoing a renaissance based on deep learning. Unique to Neuroscience, deep learning models can be used not only as a tool but interpreted as models of the brain. The central claims of recent deep learning-based models of brain circuits are that they make novel predictions about neural phenomena or shed light on the fundamental functions being optimized. We show, through the case-study of grid cells in the entorhinal-hippocampal circuit, that one may get neither. We begin by reviewing the principles of grid cell mechanism and function obtained from first-principles modeling efforts, then rigorously examine the claims of deep learning models of grid cells. Using large-scale architectural and hyperparameter sweeps and theory-driven experimentation, we demonstrate that the results of such models may be more strongly driven by particular, non-fundamental, and post-hoc implementation choices than fundamental truths about neural circuits or the loss function(s) they might optimize. We discuss why these models cannot be expected to produce accurate models of the brain without the addition of substantial amounts of inductive bias, an informal No Free Lunch result for Neuroscience. Based on first principles work, we provide hypotheses for what additional loss functions will produce grid cells more robustly. In conclusion, circumspection and transparency, together with biological knowledge, are warranted in building and interpreting deep learning models in Neuroscience.

## 1 Introduction

Over the past decade, deep learning (DL) has underpinned nearly every success story in machine learning, e.g., [57, 6] and increasingly many advances in fundamental science research, e.g., [36]. In neuroscience, DL is similarly gaining widespread adoption as an indispensable method for behavioral and neural data analysis [52, 50, 28, 43, 40, 46].

But DL offers a unique contribution to neuroscience that goes beyond its role in other fields, in that deep networks can be viewed as models of the brain. The success of DL in matching or surpassing human performance suggests it is now possible to construct models of circuits that may underlie human intelligence. As a recent review wrote, "researchers are excited by the possibility that deep neural networks may offer theories of perception, cognition and action for biological brains. This approach has the potential to radically reshape our approach to understanding neural systems" [54]. Further, DL is a democratizing force for building neural circuit models of complex function.

Here, we examine the essential claims (and promises) of DL-based models of the brain, which are that 1) Because the models are trained on a specific optimization problem, if the resulting representations

36th Conference on Neural Information Processing Systems (NeurIPS 2022).

match what has been observed in the brain, then the models reveal which optimization problem(s) the brain is solving, or 2) These models, when trained on sensible optimization problems, should generate novel predictions about the brain's representations and behavior.

These are extremely valuable potential contributions. However, given the nascent nature of such approaches and the exuberance accompanying some claims in current work, we should examine them carefully. In DL, some successes attributed to novel algorithms have been shown to instead stem from seemingly minor or unstated implementation choices [65, 21, 35]. In this paper, we ask whether Neuroscientists should similarly be cautious that DL-based models of neural circuits that make specific claims about revealing the brain's optimization functions or that generate specific neural tuning curves may tell us less about fundamental scientific truths and more about programmers' particular implementation choices, and might be more post hoc than predictive.

To explore these questions, we evaluate recent DL-based models of grid cells in the entorhinal-hippocampal circuit. The medial entorhinal cortex (MEC) and hippocampus (HPC) are part of the hippocampal formation, a brain structure critical for learning and memory. In a pair of Nobel-prize winning discoveries, HPC was shown to contain **place cells** [48], and MEC, its cortical input, was shown to contain **grid cells** [30]. Place cells each fire at one or several seemingly random locations in small and large environments [51], while grid cells fire in a spatially periodic hexagonal lattice pattern in all two-dimensional environments [30]. Over five decades, the hippocampal formation has been central to understanding how the brain organizes spatial and episodic memory, for experimentalists and theorists alike, with many mysteries remaining. A recent series of DL-based models of the circuit [15, 3, 59, 68, 47]) present a story that training neural circuits on the task of **path integration (PI)** (i.e., updating one's positional estimate by integrating velocity from self-motion), possibly with the addition of a non-negativity constraint on firing rates [59], results in the emergence of grid cells.

We use code from prior publications to demonstrate these results are due not to the core (path integration) task the network was trained to perform but to separate and specific post-hoc implementation choices that implicitly made the known tuning shapes of grid cells part of the target, even though the narrative accompanying many of these papers suggests that the emergence of those tuning curves rather naturally "falls out" from training on the core task. By leveraging theoretically-guided large-scale exploration and hypothesis-driven experimentation, we show:

1. Networks trained on path integration tasks almost always learn to optimally encode position, but almost never learn grid-like representations to do so.

2. The emergence of grid-like representations depends wholly on specifically chosen structural choices of the network and readouts, rather than on the path integration task, and the structural choices are based on the implicit goal of obtaining grid-like responses.

3. Under more-realistic structural choices for the network readouts, grid cells disappear.

4. Even with the structural choices, grid emergence can be hyperparameter and seed sensitive and non-generic.

5. Multiple grid modules, a fundamental characteristic of the grid cell system, do not emerge from path integration.

6. Grid periods and period ratios, contrary to assertions [3], are not determined by the task and are not fundamental properties that can serve as predictions about observed values in the brain; rather, they are set by hyperparameters selected by the programmer.

In short, deep learning models of MEC-HPC yield grid-like units only when a sequence of specific and biologically implausible implementation choices are made to intentionally bake grid-like units into the task-trained networks, and the emergent grid-like units lack key grid cell properties. Given the non-genericity of grid cell emergence in successfully path integrating networks, it is highly improbable that DL models of path integration would have produced grid cells as a novel prediction from task-training, had they not already been known to exist.

Moreover, it is unclear what new understanding the current models contribute, beyond or even up to what has already been shown for these circuits by existing models. Our results challenge the notion that deep networks offer a free lunch for Neuroscience in terms of discovering the brain's optimization problems or generating novel a priori predictions about single-neuron representations, and warn that caution, transparency, and biological knowledge are needed when building and interpreting such models.

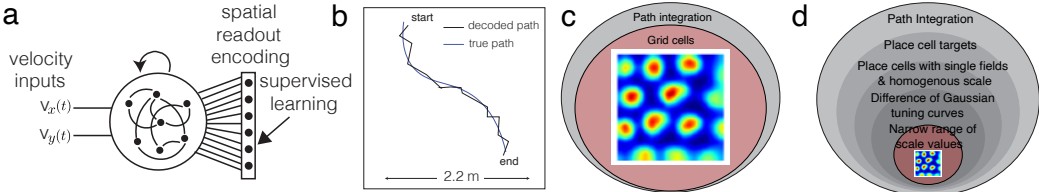

Figure 1: **Setup and claim.** (a-b) Schematic of recurrent neural network setup to predict some encoding of 2D position from 2D velocity. (b-c) Grid cells in recent DL papers, obtained in part by learning to path integrate [15, 3, 59, 60, 47], conclude that path integration creates grid cells. (d) We show that most artificial neural networks (ANNs) trained to path integrate can do so, but only a very small subset of such networks yield grid cells, and that grid cell emergence results in ANNs are post hoc: they result from post-facto selection of architectures, functions, and hyperparameter settings that specify grid cells as an implicit target.

**Code availability:** Our work benefited from previous publications' published code[47]. To facilitate further research, we similarly release ours: github.com/FieteLab/NeurIPS-2022-No-Free-Lunch.

## 2   Background: Grid Cells

Grid cells [30] are found in the medial entorhinal cortex of mammals and are tuned to represent the spatial location of the animal as it traverses 2D space. Each cell fires at every vertex of a triangular lattice that tiles the explored space, regardless of the speed and direction of movement through the space. As a population, grid cells exhibit several striking properties that provide support for a specialized and modular circuit. Grid cells form discrete modules (clusters), such that all cells within a module share a common period and orientation, while different modules express discretely different spatial periods [62]. The period ratios of successive modules have values in the range of 1.2-1.5.

The mechanism underlying grid cells is widely supported to be through attractor dynamics: Translation-invariant lateral connectivity within the grid cell network results in a linear Turing instability and pattern formation [9, 23, 7]. These models explain how grid cells can convert velocity inputs into updated spatial estimates, and make several predictions that have been confirmed in experiments, including most centrally the stability of low-dimensional cell-cell relationships regardless of environment and behavioral state, that define a toroidal attractor dynamics [24, 72, 64, 26, 25].

## 3   Experimental approach

The central messaging of existing DL models of grid cells is that training ANNs on **Path Integration (PI)** – using self-velocity estimates to track one's spatial position – causes the networks to learn grid cells [15, 3, 59, 47], even when the technical portions and code implementations of the papers involve many other critical choices without which grid cells would not emerge.

Thus, we focus on asking: if a recurrent neural network is trained on PI, what is the approximate probability that it will exhibit grid cells? We follow the setup used by many previous papers: a 2.2 m x 2.2 m arena is created, then, spatial trajectories (i.e. sequences of positions and velocities) are sampled. Networks receive as inputs the initial position and velocities, and are trained to output (some encoding of) the positions in a supervised manner (Fig. 1ab). There are multiple possible encodings of position, and, as we will show, this choice is critical. Two simple encodings are Cartesian [15] or polar [1]. Another encoding scheme is via bump functions in 2D space, with each output assigned different positions that collectively tile the space with similar tuning curve shapes [3, 59, 59, 47]. This encoding has been equated with place cells, even though place cells' fields tend to be heterogeneous in size and shape [51, 19], as well as in number [51], and, unlike the choice of a difference-of-Gaussians (DoG) or difference-of-Softmaxes (DoS) readout tuning in ANN models, do not exhibit any inhibitory surround. See Appendix A for position encoding details. For all encodings, supervised learning is used to train the network via backpropagation through time.

**Spatial tuning assessments** The spatial tuning ratemaps of hidden units in the networks are the primary basis for comparison with the brain's grid cells. To compute ratemaps, a trained network is

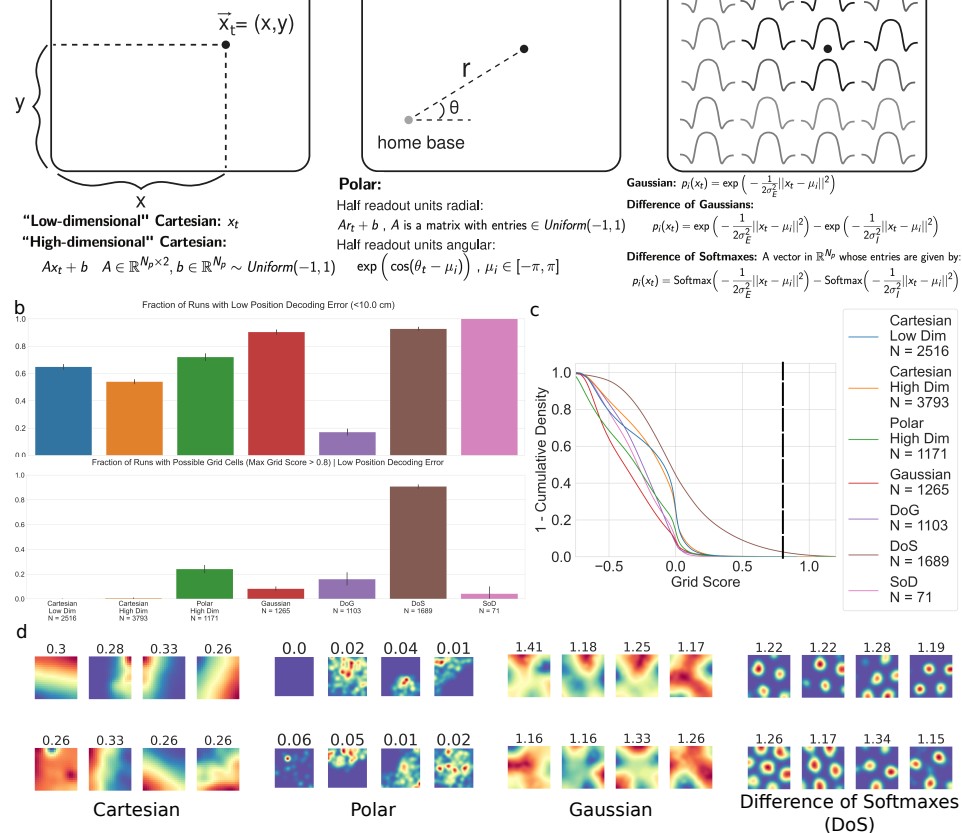

Figure 2: **Grid-like response requires highly specific target encoding.** (a) Readout encodings of spatial position. (b) Top: Across readout encodings, most networks learn the PI task. Bottom: Few networks display *possible* grid-like representations (threshold = 0.8). (c) Survival functions of grid scores per readout encoding. (d) Rate maps of topgrid-scoring units in ANNs performing good PI with i) Cartesian, ii) Polar, iii) Gaussian, iv) specifically selected (tuned) Difference-of-Softmaxes (DoS) readouts. i)-iii) do not learn grid cells. Numbers above rate maps are $60°$ grid scores.

evaluated on long trajectories that cover the 2D environment, then each hidden unit's average activity per spatial bin is computed. Ratemaps of units are compared with grid cells through the gridness score use by [3, 59, 60, 47]. *We are extremely lenient with classifying a particular network training run a success: if even a single hidden unit has a grid score above a certain threshold, we say the model possibly possesses grid cells.* The grid score, when applied to ANN units without additional criteria, is not perfect since cells classified by grid scores represent only an upper bound on the total number of grid cells (e.g. the high grid score given to units with triangular symmetry without a periodic pattern, Fig. 2d (Gaussian readouts)); for details, see Appendices B and C.

## 4 Networks trained on path integration tasks learn to estimate position, but rarely learn grid cells

We demonstrate that most path-integrating networks do not converge to a grid-like solution, instead requiring very specific architectural choices including readout tuning functions. Grid-like representations emerge when the programmer makes choices that, rather than relating to the path integration objective or biologically realistic place cells, are designed post-hoc to produce grid cells.

We ran large-scale hyperparameter sweeps across common implementation choices: 1) Architectures: RNN [20]; LSTM [33]; GRU [13]; UGRNN [14]; 2) Activation: Sigmoid; Tanh; ReLU; Linear; 3) Optimizers: SGD, Adam [41]; RMSProp [32] 4) Supervised Targets: Cartesian; Polar; high-dimensional bump-like readout population code with Gaussian [3], Difference-of-Softmaxes (DoS)

[59, 60, 47] or Difference-of-Gaussians (DoG) tuning curves. 5) Loss: mean squared error on the agent's Cartesian position [37, 15]; geodesic distance on the agent's polar position [1]; cross entropy on a high-dimensional population of bump-like readout units [3, 59, 47] 6) Miscellaneous: recurrent & readout dropout, initialization, parameter L2 regularization, seed.

For networks with bump-like population readouts, we additionally swept: 1) Width $\sigma$ of Gaussian readouts; 2) Whether the bump-like readouts have homogeneous or heterogeneous field widths; 3) For DoG or DoS readouts, the surround scale $s$, i.e., the ratio between the inhibitory and excitatory Gaussian standard deviations ($s \stackrel{\text{def}}{=} \sigma_I/\sigma_E$); 4) For DoG readouts, the ratio of amplitudes $\alpha_E/\alpha_I$ between the two Gaussians. 5) Number of fields per readout unit. Evaluating the entire hyperparameter volume is computationally prohibitive, so we evaluated a subvolume most consistent with previous papers, focusing our exploration around conditions that did produce grid cells. In this sense, our search was biased toward configurations shown to produce grid cell emergence and thus our findings about the fragility of these solutions conservatively favored these solutions as much as possible. All sweeps are provided in Appendix F.

To evaluate whether a network learns to optimally estimate spatial position from velocity inputs, we measured its position decoding error using previous papers' methods [3, 59, 47]: using the network's output Cartesian positions (if trained on Cartesian targets) or by decoding position from the network's outputs. Any network with error $< 10$ cm was considered to have achieved optimal position encoding.

In total, we trained $> 11,000$ networks and found that most succeed in learning to path integrate (Fig. 2a, Top), but few learn grid cell representations (Fig. 2a, Bottom). This is consistent with earlier work [37, 1] demonstrating that networks can learn to path integrate and solve other hard navigational problems (e.g. self-localization across multiple environments and identification of spatial environment from ambiguous cues, a case of self localization and mapping or SLAM) without grid-like units emerging as a solution.

## 5 Grid-like unit emergence requires specific supervised target functions

We next sought to characterize when grid cells are learnt under different encodings of 2D spatial position in the readout units (i.e. supervised targets). We tested multiple encodings: i) Cartesian, ii) Polar, iii) Gaussian, iv) Difference-of-Gaussians (DoG), and v) Difference-of-Softmaxes (DoS).

We found we that in the ANN network architecture of Fig. 1a, grid cells do not emerge from Cartesian or Polar readouts, consistent with earlier work [37]. Similarly, they do not emerge from Gaussian encodings (Fig. 2) [37]. Consistent with this result, the Gaussian readouts of [3] used in tandem with 50% dropout and a different architecture do not yield (square or hexagonal) grid cells without dropout (result not shown). and, as shown recently and independently by [70], although grid-like responses can be obtained with Gaussian readouts after the addition of another constraint, they disappear without. [59, 60] critiqued [3] to show that the hexagonal patterning of cells in [3] was indistinguishable from low-pass filtered noise. However, their [59, 60] focus was to argue that a neural nonlinearity (which they termed a non-negativity constraint, though any non-odd function suffices) robustly produces hexagonal firing – which they showed by replacing the simpler Gaussian-like readouts of [3] with a specific DoG/DoS readout.

We found that DoG/DoS readouts [59, 60] were critical for producing grids (Fig. 2), reproducing the main results of [59], even with non-negativity constraints: Lattices of any geometry (hexagonal or square) only emerge with DoS fields, corresponding to a small and particular subset of DoG fields.

**Grid period values set by hyperparameters, and multiple modules do not emerge.** Next, two prominent features of grid cells are their intrinsically set periods (invariant to the external environment) and the existence of a discrete set of grid periods that scale by a rough factor of $1.4$ between adjacent scales [62]. Multi-periodicity is critical for unambiguous spatial coding over large scales. We asked whether ANN models generate multiple periods and whether their the period values are fundamental or hyperparameter dependent.

To ensure we would obtain at least some grid cells, we fixed the readouts to be DoS, and swept over different scales $\sigma_E$ of the DoS. We found that almost all runs had a unimodal distribution of grid periods (Fig. 3a), meaning the networks learnt only one module of grid cells. Contemporaneously with the NeurIPS review process, other researchers independently reported the same result [56].

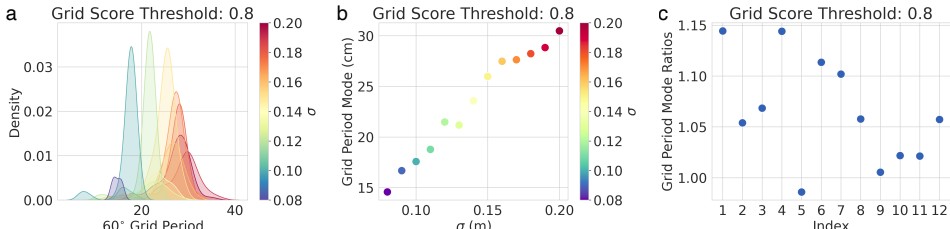

Figure 3: **The spatial scale of grids is set by hyperparameters and multiple modules do not emerge.** (a) Over a wide sweep of DoS (Fig. 2a bottom) target field widths $\sigma_E$, the distribution of grid periods is unimodal (each color: period distribution from 3 runs with same $\sigma_E$ value; different periods are only obtained by varying $\sigma_E$), meaning multiple grid modules do not emerge, in contrast to the brain's grid cell circuit. (b) The chosen target field width $\sigma_E$ determines the grid period mode, meaning that grid period is not a prediction of the models. (c) If we smoothly sweep $\sigma_E$ as a proxy for simulating different modules, the distribution of adjacent periods produces ratios closer to 1 than the experimental ratios of $\sim 1.4$.

Further, we found that the period of formed grid-like representation is completely determined by the width $\sigma_E$ of the externally imposed readout DoS (Fig. 3) and other specific parameter choices. The period of the grid-like responses increased monotonically with the width of the DoS readout (Fig. 3b). Since individual networks did not learn multiple modules, we used the somewhat discrete distribution of peaks of the single module formed when sweeping the DoS $\sigma_E$ more continuously to compute grid period ratios. These period ratios from adjacent peaks led to non-biological values (Fig. 3c).

**Other details with DoG/DoS readouts affect grid emergence.** In all DL-grid cell papers we examined [3, 15, 59, 60, 47], we discovered implementation details critical to the emergence of grid cells that were not stressed in the claims. As one example, we discovered an implementation detail essential for the emergence of grid cells in a series of papers [59, 60, 47] that is unmentioned in main texts and supplements. These papers report using an unnormalized equi-norm Difference-of-Gaussian (DoG) readout target function ([59] Appendix C1,[60] Methods 4.2, [47] Appendix C1), i.e. a DoG with amplitude parameters set to 1, but their code uses a Difference-of-Softmaxes (DoS) target function. When we trained ideal grid-forming ReLU networks with equi-norm DoG tuning curves, sweeping the receptive field $\sigma$ and surround scale $s$, they did not result in grids (Fig. 4b). The Fourier analysis (below) explains why equinorm-DoG tuning should not produce lattices (Fig. 4c).

We next trained ReLU RNNs on general DoG readouts, sweeping the component amplitudes $\alpha_E, \alpha_I$ while holding $\sigma_E, \sigma_I$ fixed at ideal values. The theory of [59] predicts that outside the feasible region (blue boxed region of Fig. 4c), no lattices should emerge, but inside the feasible region, all/most RNNs should learn grids. We found the first prediction held, but the second did not: most display grid score distributions comparable to or worse than low-pass-filtered-then-thresholded noise, and well below the ideal-width DoS grid score distribution (Fig. 4d). To investigate, we swept densely inside the feasible region, additionally matching the amplitudes created by DoS (Fig. 4c; Fig. 10); one run out of 1086 surpassed the DoS grid score distribution ($\alpha_E/\alpha_I \approx 3.5714$, seed=1), but its two "sibling" runs (all hyperparameters same; seeds: 0, 2), Fig. 4e. Thus, and contrary to [59]'s theory based on static function-fitting, DoG readouts at most rarely produce lattices when implemented in actual RNN simulations.

**Fourier analysis of Turing instability provides intuition for the preceding empirical results.** Why do only Difference-of-Softmaxes (DoS) or very specific DoG readouts produce grid-like units? We shed light on our present findings by restating the essence of previous analyses of first-principles models [9, 38] here, and leveraging the connection made between these models and trained deep networks in [59]. In the first-principles continuous attractor models, neural dynamics are given by $\dot{r}(x) = -r(x) + g(W \star r)$, where $x$ designates the neural index (in a continuum approximation for neurons), $W \star r$ designates the total (integrated) inputs from the network to the neuron at index $x$, and $g$ is the non-linearity, if the recurrent weight interaction is translationally invariant, then $W(x, x') = W(x - x') = W(\Delta x)$. Under DoG interactions:

$$W(\Delta x) \equiv f(\Delta x) = \alpha_E \exp\left(-\frac{(\Delta x)^2}{2\sigma_E^2}\right) - \alpha_I \exp\left(-\frac{(\Delta x)^2}{2\sigma_I^2}\right) \tag{1}$$

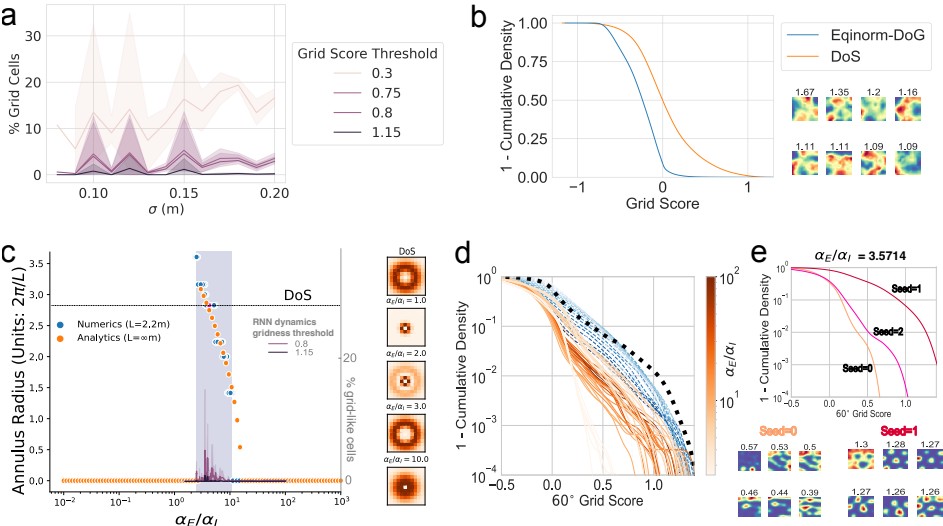

Figure 4: **Other details with DoG/DoS readouts affect grid emergence:** Even with DoS/DoG translation-invariant targets, grid solutions are sensitive: (a) Modestly varying the DoS width $\sigma_E$ can cause grids to disappear. (b) Left: Grid scores of networks trained with equinorm-DoG versus DoS readouts [59, 47, 60] shows DoS is critical for high grid scores. Right: Rate maps of top grid scoring units from equinorm-DoG networks. (c) A necessary condition for grid emergence with DoG readouts is that the Fourier transform of the readout correlation matrix contain an annulus of radius $> 2\pi/L$ ($L$ is the size of the enclosure in which the RNN is trained). The hyperparameter region where this condition is met is $< 1$ order of magnitude (FT annulus radius computed analytically (orange) and through numerical construction of fields in finite environment (blue)). RNNs do not learn periodic responses outside this region, but most RNNs inside the region do not either, meaning the annular radius criterion is insufficient. DoS readouts are similar to one particular choice of DoG amplitudes, and only DoG amplitudes very close to the DoS point succeed in producing periodic responses. (d) Across $\alpha_E/\alpha_I$ DoG ratios, DoG networks generally score worse than filtered-and-thresholded noise (blue) and worse than DoS (black). (e) More densely sweeping within the theoretically feasible $\alpha_E/\alpha_I$ DoG region, and choosing $\alpha_E$ and $\alpha_I$ magnitudes closely matching DoS, shown in (c) still showed only one ratio that did at least as well as a DoS (App. E), but this result was true for one out of three seeds; two "sibling" runs with otherwise identical settings produced poor gridness.

where $\Delta x$ refers to the difference of indices between the neural pair linked by the weights. The evolution of activity can be decomposed into the growth and decay of Fourier components of the rate vector, which is fully determined by the Fourier transform of $W$, which is given by:

$$\tilde{f}(k) = \int_{\mathcal{R}} d(\Delta x) f(\Delta x) e^{ik\Delta x} = \alpha_E \sigma_E \exp\left(-\frac{\sigma_E^2 k^2}{2}\right) - \alpha_I \sigma_I \exp\left(-\frac{\sigma_I^2 k^2}{2}\right) \qquad (2)$$

Here $\alpha_E$ ($\alpha_I$) denotes the strength and $\sigma_E$ ($\sigma_I$) denotes the scale of excitation (inhibition). For linearized dynamics that approximate $\dot{r}(x) \sim -r(x) + f(\Delta x) \star r$ (i.e., $g$ has been linearized), the solution will be periodic if the maxima of $\tilde{f}(k)$, given by $[k^*]^2 = \frac{2}{\sigma_E^2 - \sigma_I^2} \log\left(\alpha_E \sigma_E^3 / \alpha_I \sigma_I^3\right)$, contains sufficient power and if $k^* \neq 0$. Specifically, the condition for pattern formation is $\tilde{f}(k^*) > 1$ [10, 38]. In particular, the inhibitory surround contained in $f(\Delta x)$, with strength $\sigma_I$, is key to pattern formation; if $\sigma_I \to \infty$ or $\alpha_I \to 0$, the maximum of the Fourier-transformed weights is at the origin ($k^* = 0$), corresponding to a DC (non-patterned/non-periodic) activity state. Similarly, only particular choices of the ratio $\alpha_E/\alpha_I$ work, Fig. 4c. In sum, a Gaussian recurrent interaction cannot produce periodic patterns in the continuum limit (relatively large number of cells, large environment), as known from first principles models.

The theory of [59] contributes a connection between these first principles models and feedforward networks performing supervised least-squares regression ("function approximation framework") onto a target readout $P$, through the observation that gradient optimization of the MSE reconstruction loss $||P - W_{\text{readout}} r||^2$ can be approximated as $\dot{r} = -\lambda r + \Sigma r$, where $\Sigma_{x,x'} = \sum_i P_i(x) P_i(x')^T$ is the

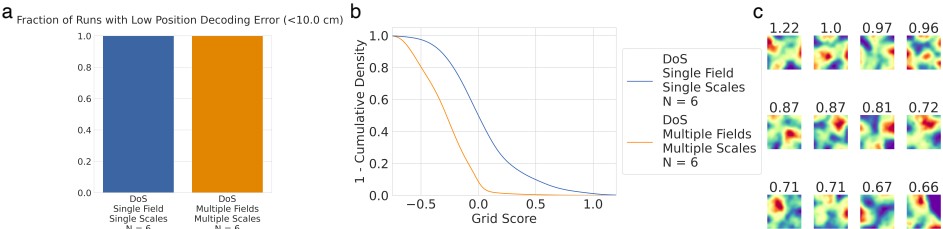

Figure 5: **Adding place cell-like heterogeneity to readouts prevents grid emergence.** We selected DoS RNNs with the best hyperparameters for grid cell emergence (RNN or UGRNN, ReLU, $\sigma_E = 0.12$ cm, $s = 2.0$, 3 seeds), then tested the effect of multiple fields per place cell ($\sim 1 + \text{Pois}(3.0)$) and multiple scales (receptive field width $\sigma_E \sim \text{Unif}(0.06, 1.0)$ m and surround scale $s \sim \text{Unif}(1.25, 4.5)$). (a) Networks with multi-scale multi-field DoS readouts all obtain low position decoding error. (b) Multi-scale multi-field DoS readouts do not learn grid cells. (c) Highest-scoring rate maps from multi-field multi-scale networks.

specified spatial correlation matrix of the target readouts and $\lambda$ is a regularization parameter. This dynamics now resembles that of first-principles grid cell models, provided the readout correlation matrix has the same form as the first-principles recurrent interaction matrix: $W(x, x') = \Sigma_{xx'}$, upto scaling factors. If the readout target functions are set to be translationally invariant with DoG tuning curves, the readout correlation matrix is a difference of multiple Gaussians (Appendix D), Fig 4c. By the linear stability analysis outlined above, it follows that DoG tuning can sometimes produce grids, but simple Gaussian tuning curves will not generically produce periodic patterns without additional assumptions. This is true whether we consider discretized real and Fourier space or take the continuum limit in both cases: Gaussian readouts generate roughly uniform (non-periodic) activation as the dominant state. Only under the further assumptions of discretization induced by a small spatial environment, higher-order non-linear effects, and orthogonality of hidden units, might Gaussians be theoretically predicted to produce periodic responses [59], and then the periods and shapes of the grids will depend on the environment properties. However, neither hexagonal nor square grids emerge at a significance level beyond filtered and thresholded noise in actual RNNs trained to PI with Gaussian and most DoG readouts.

## 6 Grid cells disappear with realistic readout population heterogeneity

Place cells, to which grid cells project, differ significantly from the highly idealized single-scale, single-field translation-invariant ANN readouts, even ignoring the particular center-surround shape for each field. Place cells have heterogeneous field widths, many with multiple fields [51, 19] and non-uniform spatial correlations. Place cells at similar dorsoventral locations can exhibit a range of field sizes, and single place cells themselves exhibit a diversity of field widths [19]. This naturally leads to the question: Will readout targets with more place cell-like heterogeneous responses still produce grid cells? We found that networks trained with multiple-field, multiple-scale DoS readouts achieve position decoding error as low as single-field single-scale DoS encodings (Fig. 5a), but do not learn grid cells (Fig. 5bc). This finding is consistent with the strong requirement in ANN models of a translation-invariant readout code for grid emergence [3, 59]. Translation invariance is a specific property of grid cells, but it is not likely true of place cells, which as a population over-represent borders, landmarks and reward locations [53, 49, 69, 31, 34, 18, 16, 73, 27, 5]. These observations further demonstrate that ANN models essentially build the known structure of grid cells into their targets, rather than obtaining them from training on simple tasks with plausible readouts.

## 7 Why path-integrating ANNs might achieve high predictivity of MEC data

We conclude by introducing a puzzle. A recent NeurIPS spotlight [47] notes that networks trained on single-field single-scale DoS readout encodings explain variance in mouse MEC neural activity at nearly 100% of variance explained by other mice. In contrast, our results demonstrate that these networks learn few grid cells, produce unimodal grid period distributions inconsistent with biological

grid cells, and require readout encodings inconsistent with biological place cells to do so. How are these networks able to predict mouse MEC neural activity so well?

The analysis code is not open source, so we are unable to investigate this puzzle. However, we offer a conjecture with preliminary evidence. The analysis of [47] linearly regressed rate maps from one agent (mouse or network) onto rate maps from another mouse, and used Pearson correlation as a measure of "neural predictivity." We conjecture that different architecture-activation pairs achieve different neural predictivity scores because different pairs learn different intrinsic dimensionalities that then provide richer/poorer bases for linear regressions.

To explore our conjecture, we trained [47]'s 5 architectures (RNN, LSTM, GRU, UGRNN, SRNN) with DoS readouts and ReLU activations (5 seeds per architecture-activation pair). For each trained network, we extracted rate maps and computed a standard linear measure of dimensionality called participation ratio [42], then plotted each pair's average participation ratio against the published neural predictivity. We found networks with higher dimensional rate maps have higher neural predictivity (Fig. 6).

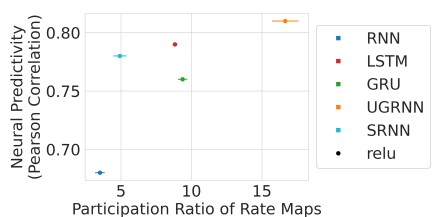

Figure 6: **Networks with higher (lower) dimensional rate maps display higher (lower) "neural predictivity" of mouse MEC rate maps.** Each dot is an architecture-activation pair; participation ratio averaged over 5 seeds. Neural predictivity from [47].

We caution this correlation between dimensionality and neural predictivity is not (yet) strong evidence. However, we predicted that similar results would be found for other species and modalities, and during the subsequent NeurIPS review process, two independent research groups confirmed our predictions in macaque vision [2] and human audition [66]. If correct, the conjecture raises the interesting research question of whether linear regression-based comparisons of ANNs with neural data might produce better matches to biology more because ANNs have higher dimensional representations than competitors, than because of any detailed similarity.

## 8   Discussion

For research that uses deep networks as models of the brain, there is a fundamental obstacle to making the claim that a given optimization problem is what the brain is solving: If we know the responses of a significant fraction of units from biological networks performing a certain task, we cannot infer the loss function that the brain is optimizing since in principle, numerous different loss functions can have the same/similar minima (Fig. 7 top). In other words, there is typically a *many-to-one* mapping between loss functions and some point in model space where the losses have a minimum. Some of the different grid models from DL and first principles show that this is possible [15, 3, 59, 68, 9]. Conversely, given a reasonable optimization problem that we select based on an organism's ecological niche, we cannot infer a single solution (and thus build truly predictive single-cell tuning models), since there exist several minima to that optimization problem (Fig. 7 bottom). In other words, there is typically a *one-to-many* mapping from a loss function to its set of solutions.

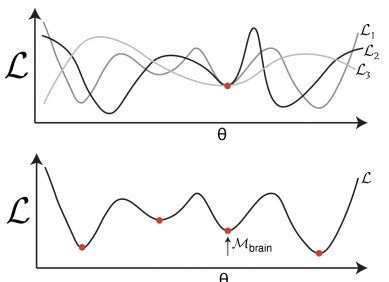

Figure 7: **Challenges in achieving the two central claims of recent DL models of neuroscience**: Top: Building a model that replicates observed neural responses does not guarantee that the loss function used is the brain's objective, as multiple objectives can share a solution. Bottom: Training a network on a plausible loss function or even the correct loss function need not yield the solution the brain has selected because the loss function may have multiple minima, of which the brain selects one based on its constraints, while an ANN selects another, based on the optimization technique used.

To break this uniqueness problem and arrive at truly predictive models and a better understanding of the brain's optimization problems, we must understand the specific inductive biases and constraints present in the biological system we are trying

to model. It is untenable to expect success without doing so. This is what we refer to as an informal neural 'No Free Lunch'.

Can we learn about brain circuits from DL models *without* considering biological inductive biases? Low-dimensional latent representations and dynamics that emerge as necessary for solving difficult problems are possibly sufficiently robust and abstract to be predictive of populations in a neural circuit. For instance, any model solving the task of finding ripe apples in color photos, will create some abstracted representation of round red objects; this would also be a robust prediction for neural systems, but not unique to them. On the other hand, we should generally not expect detailed single neuron tuning correspondences without specific additional constraints or inductive biases: If a low-dimensional latent representation is necessary to solve a task, there are a multiplicity of ways to project it onto the activities of a large number of neurons. Which projection the brain selects depends on factors including energetics, neuron number, downstream uses, and the vagaries of evolutionary dynamics; the projections of ANNs depend on similar factors but specific to the ANN's loss and the vagaries of gradient descent learning. Consistent with this, models of the visual pathway [71, 4] and circuits that solve latent inference tasks [63, 55, 1, 67] exhibit population-level representations of abstract variables necessary to solve the task. By contrast, DL-based grid cell models make fine-grained claims about single-neuron tuning, which should be surprising without the incorporation of significant additional constraints. Only in cases where task constraints completely overwhelm all system-specific constraints, might we expect the natural emergence of alignment at the single-unit level.

Returning to grid cells, since they do not generically arise in networks trained to path integrate, path integration is not a sufficiently constraining task. Theoretical work on grid cell representations [22, 61, 45] suggests additional critical features of the code: an exponentially large coding range and robustness/intrinsic error correction, both of which translate into the problem of packing and maximally separating a large set of coding states into a compact space [61]. We hypothesize that the following key properties of the grid cell code may form a biologically relevant sufficient set for their emergence: 1) non-negative activity; 2) path integrating (PI) code that is translation invariant [23, 9, 12]; 3) exponential representational capacity [8, 61, 44]; 4) intrinsic error-correcting capabilities [8, 61]; and 5) uniformly distributed (whitened) information across cells. Several of these are general properties of neural codes, and could increase the ability of ANN models to make de novo rather than post hoc predictions. For recent promising work, see [17].

First-principles continuous attractor network (CAN) models of grid cells made several novel predictions subsequently confirmed in experiments [39]: the invariance of cell-cell relationships across environments and behavioral states (constituting, in the terminology of machine learning, "far out of distribution" predictions) that define an invariant toroidal attractor manifold [72, 64, 26, 25], grid-like patterning in the cortical sheet [29], and many others that remain to be tested. Deep learning-based models should be held to similar standards.

In sum, ANN models of the brain that reproduce specific tuning curves should not center their claimed achievement on producing the curves if these are used as implicit or explicit parts of the training target (given the expressive power of deep networks, it is no revelation that training them to generate a given tuning will in fact succeed), but rather should characterize the conditions under which the particular tuning does and does not emerge, to consider which inductive biases are critical, and to explicitly state what principles and de novo predictions can be extracted from the models.

# 9   Acknowledgements

This work was supported by the ONR, the NSF, the Simons Foundation through the SSCGB program, and the HHMI through the Faculty Scholars Program. MK was supported by the MathWorks Science Fellowship.

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
