# A    Position Encodings

Suppose we sample a sequence of positions $x_0, ..., x_T \in \mathbb{R}^2$ and a sequence of velocities $v_1, ..., v_T \in \mathbb{R}^2$, where $x_t = x_{t-1} + v_t$. We want to train the networks in a supervised manner to predict (a possible encoding of) their position. We used the below encodings as different regression targets. For some encodings that required place cell populations, we denote the number of place cells $N_p$ and denote their locations $\{p_i\}_{i=1}^{N_p}$, sampled uniformly at random within the 2.2 m $\times$ 2.2 m arena.

- **"Low-dimensional" Cartesian:** $x_t$
- **"High-dimensional" Cartesian:** Let $A \in \mathbb{R}^{N_p \times 2}, b \in \mathbb{R}^{N_p}$ have entries sampled i.i.d. from $Uniform(-1, 1)$. The target is a vector in $\mathbb{R}^{N_p}$ given by:

$$Ax_t + b$$

- **Polar:** Let $(r_t, \theta_t)$ denote the polar form of the agent's position $x_t$. The target is a vector in $\mathbb{R}^{N_p}$, half comprised of "distance" units and half comprised of "direction" units. Let $A \in \mathbb{R}^{0.5N_p \times 1}, b \in \mathbb{R}^{0.5N_p}$ have entries sampled i.i.d. from $Uniform(-1, 1)$; the distance cells have activites:

$$Ar_t + b$$

  Let $\mu \in [-\pi, \pi]^{N_p/2}$ have entries sampled i.i.d. uniformly at random. The direction cells have entries given by von-Mises-like bumps:

$$\exp\Big(\cos(\theta_t - \mu_i)\Big)$$

- **Gaussian:** A vector in $\mathbb{R}^{N_p}$ whose entries are given by:

$$\exp\Big(-\frac{1}{2\sigma_E^2}||x_t - p_i||^2\Big)$$

- **equi-norm Difference of Gaussians:** A vector in $\mathbb{R}^{N_p}$ whose entries are given by:

$$\exp\Big(-\frac{1}{2\sigma_E^2}||x_t - p_i||^2\Big) - \exp\Big(-\frac{1}{2\sigma_I^2}||x_t - p_i||^2\Big)$$

- **Difference of Softmaxes:** A vector in $\mathbb{R}^{N_p}$ whose entries are given by:

$$Softmax\Big(-\frac{1}{2\sigma_E^2}||x_t - p_i||^2\Big) - Softmax\Big(-\frac{1}{2\sigma_I^2}||x_t - p_i||^2\Big)$$

- **Softmax of Differences:** A vector in $\mathbb{R}^{N_p}$ whose entries are given by:

$$Softmax\Big(-\frac{1}{2\sigma_E^2}||x_t - p_i||^2 + \frac{1}{2\sigma_I^2}||x_t - p_i||^2\Big)$$

# B    Grid Scores and Grid Cell Thresholds

What qualifies as a grid cell? The most commonly used method of quantifying grid cells is via the "grid score", which functions by binning neural activity into rate maps using spatial position, applying an adaptive smoother, then taking a circular sample of the autocorrelation centered on the central peak and comparing it to rotated versions of the same circular sample. The 60° grid score is specifically given by:

$$(corr[60] + corr[120])/2 - (corr[30] + corr[90] + corr[150])/3$$

We used the same grid scorer implementation used by [3] (https://github.com/deepmind/grid-cells/blob/master/scores.py), [59] (https://github.com/ganguli-lab/grid-pattern-formation/blob/master/scores.py) and [47].

What score is sufficient to qualify as a grid score? Experimentalists have used thresholds of 0.3 [58] and 0.349 [11] on biological neurons, whereas computationalists have used 0.3 [47] and 0.37 [3, 59] on artificial neurons. We found that for artificial neurons, these thresholds are far too low (Fig. 8); ANN units with grid scores $> 0.4$, and even as high as 1.3, often look nothing like grid cells. This is because the grid score looks for 60° rotational symmetry, and while grid cells are indeed symmetric, so are many other rate maps.

## C   Number of Bins for Computing Rate Maps

The first step in computing grid scores is determining the number of bins to use to compute rate maps. The original experimental work used 5 cm x 5 cm bins [30]. Since the square arena used in these experiments is 2.2 m by 2.2 m (same as [3, 59, 47]), the number of bins should be 44 x 44. Due to inconsistencies in the number of bins previously used, we checked what effect, if any, the number of bins has on the distribution of grid scores. We found that 20x20, 32x32 and 44x44 appears to have little to no differences (Fig. 9), so we chose to use 44 x 44 bins to be consistent with experimentalists.

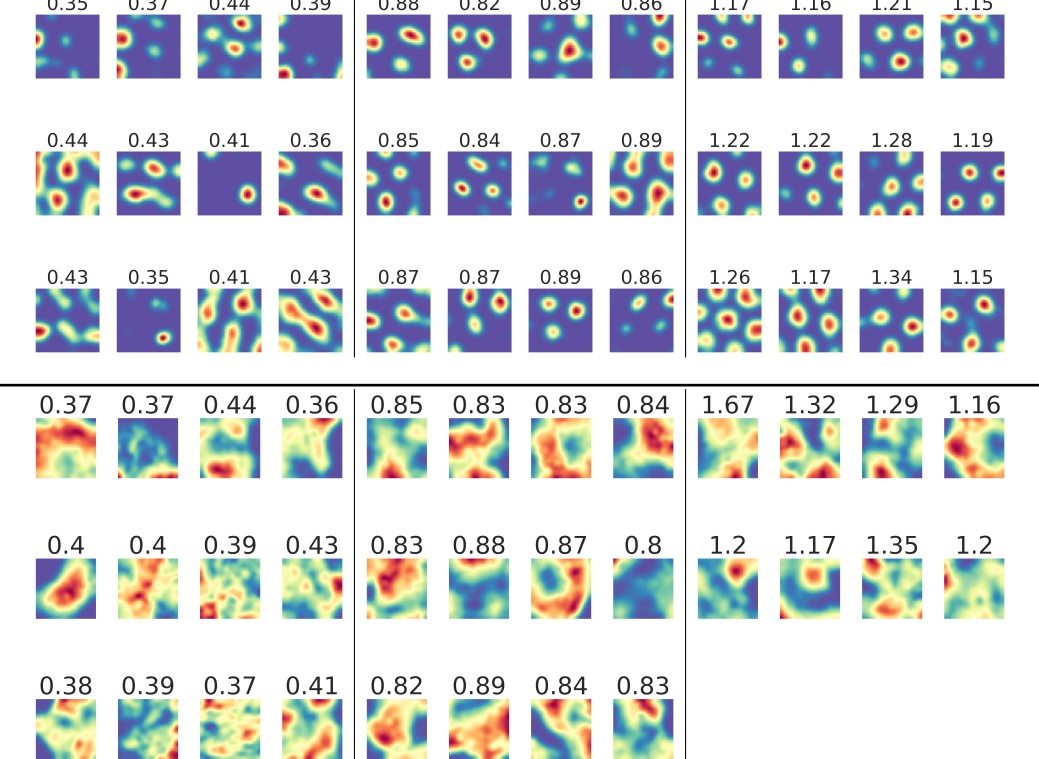

Figure 8: **Grid scoring is the dominant method to identify grid cells, but we found it performs inadequately at identifying grid cells.** Top: example rate maps from low position decoding error Difference-of-Softmaxes (DoS) networks three grid score ranges: $[0.35, 0.45), [0.8, 0.9), [1.15, \infty)$. Bottom: example rate maps from low position decoding error Difference-of-Gaussians (DoG) networks. We considered three grid score thresholds: 0.3 (used by some experimentalists), 0.8 (low probability of finding grid cells), 1.15 (decent probability of finding grid cells).

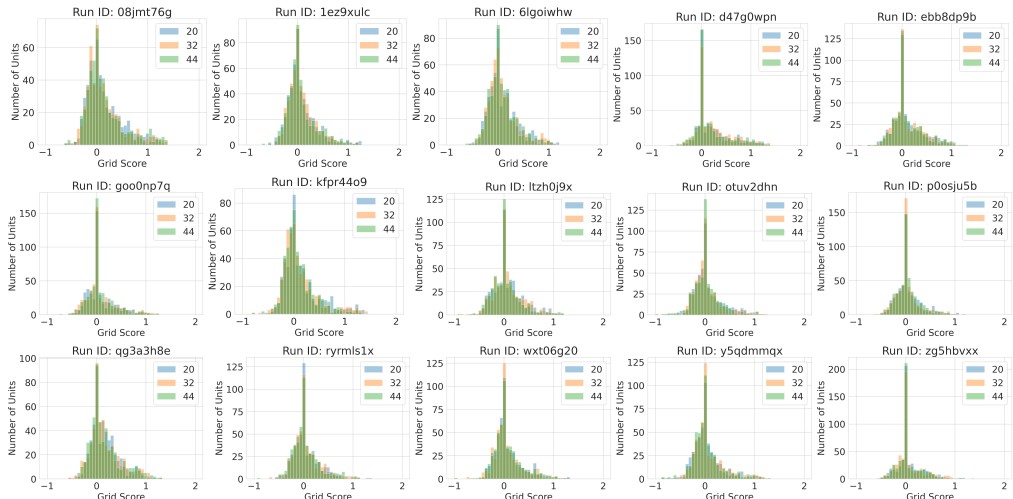

Figure 9: Grid score distributions do not differ as a function of number of bins: 400 (20 x 20; blue), 1024 (32 x 32; orange), 1936 (44 x 44; green).

## D  Place cell spatial autocorrelation

In this section, we will derive the form of the place cell correlation function. When the spatial tuning curve is a difference-of-Gaussians, the correlation function is also a Difference-of-Gaussians, albeit with different parameters. This calculation is performed in d dimensions. In simulations, $d = 2$. Consider the spatial tuning curve:

$$P(\mathbf{x}; \vec{\mu}) = \alpha_E \exp\left(-\frac{(\mathbf{x} - \vec{\mu})^2}{2\sigma_E^2}\right) - \alpha_I \exp\left(-\frac{(\mathbf{x} - \vec{\mu})^2}{2\sigma_I^2}\right)$$

So the correlation function of $\mathbf{x_1}$ and $\mathbf{x_2}$ with $\mathbf{\Delta x} = \mathbf{x_1} - \mathbf{x_2}$ (assuming that place cell centers are distributed isotropically and can be integrated over) is given by:

$$C_{\mathbf{x_1 x_2}} = \int P(\mathbf{x_1}; \mu) P(\mathbf{x_2}; \mu) d\mu$$

Simplifying the above expression, using the identity: $\int d\mathbf{x} \mathcal{G}(\mathbf{x}; \mu_f, \sigma_f) \mathcal{G}(\mathbf{x}; \mu_g, \sigma_g) = \left(\frac{\sqrt{2\pi}\sigma_f \sigma_g}{\sqrt{(\sigma_f^2 + \sigma_g^2)}}\right)^d \mathcal{G}(\mathbf{r}; \mu_f - \mu_g, \sqrt{\sigma_f^2 + \sigma_g^2})$, where $\mathcal{G}(\mathbf{x}; \mu, \sigma) = \exp\left(-(\mathbf{x} - \mu)^2/2\sigma^2\right)$

$$C_{\mathbf{x_1 x_2}} = \alpha_E^2 \left(\sqrt{\pi}\sigma_E\right)^d \exp\left(\frac{-\mathbf{\Delta x}^2}{4\sigma_E^2}\right) + \alpha_I^2 \left(\sqrt{\pi}\sigma_I\right)^d \exp\left(\frac{-\mathbf{\Delta x}^2}{4\sigma_I^2}\right) - 2\alpha_E \alpha_I \left(\frac{\sqrt{2\pi}\sigma_E \sigma_I}{\sqrt{(\sigma_E^2 + \sigma_I^2)}}\right)^d \exp\left(\frac{-\mathbf{\Delta x}^2}{2(\sigma_E^2 + \sigma_I^2)}\right)$$

The Fourier transform of this quantity, $\tilde{C}$ can be calculated using the identity

$$\int_{R^d} \exp(-\mathbf{x}^2/2\sigma^2) \exp(i\mathbf{k} \cdot \mathbf{x}) dx = (\sqrt{2\pi}\sigma)^d \exp(-\mathbf{k}^2 \sigma^2/2)$$

$$\tilde{C}(k) = \alpha_E^2 \left(\sqrt{2}\pi\sigma_E^2\right)^d \exp\left(-\mathbf{k}^2 \sigma_E^2\right) + \alpha_I^2 \left(\sqrt{2}\pi\sigma_I^2\right)^d \exp\left(-\mathbf{k}^2 \sigma_I^2\right) - 2^{d/2+1}\alpha_E \alpha_I \left(\pi\sigma_E \sigma_I\right)^d \exp\left(-\mathbf{k}^2(\sigma_E^2 + \sigma_I^2)/2\right)$$

(3)

Using this equation, we numerically solve for the maximum $k^*$.

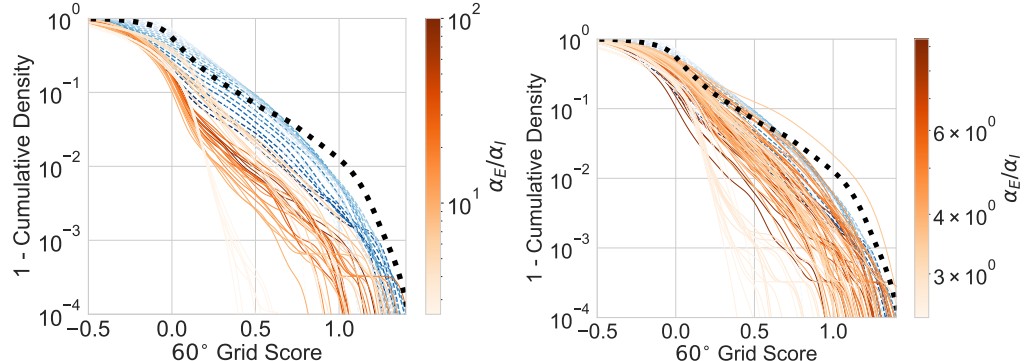

Figure 10: Left: Fig. 4 from main text. A broad sweep of DoG hyperparameters show that DoG (red) obtains lower grid score distributions than filtered-and-thresholded noise (blue) and than Difference-of-Softmaxes (black). Right: A narrow sweep of DoG hyperparameters in the "feasible region" of Fig. 4c show that almost all DoG (red) obtains lower or comparable grid score distributions than filtered-and-thresholded noise (blue), and lower than Difference-of-Softmaxes (black). Note: Colorbars are different between left and right.

## E  DoG and Narrow DoG Sweeps

We evaluated non-equinorm Difference-of-Gaussian readouts, with activations given by:

$$p_i(x) = \alpha_E \exp\Big(-\frac{1}{2\sigma_E^2}||x-\mu_i||^2\Big) - \alpha_I \exp\Big(-\frac{1}{2\sigma_E^I}||x-\mu_i||^2\Big)$$

The analytics tell us that the ratio $\alpha_E/\alpha_I$ should determine whether the Fourier annulus radius $> 1$ and thus whether grid-like representations might emerge. Over a broad sweep of $\alpha_E/\alpha_I$ ratio values, we found that all networks' grid score distributions were dominated by filtered-then-ReLU-thresholded noise $\sim Uniform(-1, 1)$, which were in turn dominated by ideal-width Difference-of-Softmaxes readouts (Fig. 10). This shows that most DoG readouts will fail to produce lattices.

Next, we ran a much narrower sweep of $\alpha_E/\alpha_I$ ratio values in the range $(2.5, 10)$, which [59] predicts should robustly yield lattices. We found that most networks' grid score distributions were dominated by or indistinguishable from filtered-then-ReLU-thresholded noise $\sim Uniform(-1, 1)$, which were in turn dominated by ideal-width Difference-of-Softmaxes readouts (Fig. 10). One ratio $\alpha_E/\alpha_I = 3.5417$ appeared to outperform DoS, but this was due to a single lucky seed (Fig. 4e). This shows that, even when theory predicts the formation of lattices, most DoG readouts will fail to produce lattices.

# F Sweeps

## F.1 Cartesian (Low Dimensional)

```
method: grid
metric:
  goal: minimize
  name: pos_decoding_err
parameters:
  Ng:
    values:
      - 1024
  Np:
    values:
      - 2
  activation:
    values:
      - relu
      - tanh
      - sigmoid
      - linear
  batch_size:
    values:
      - 200
  bin_side_in_m:
    values:
      - 0.05
  box_height_in_m:
    values:
      - 2.2
  box_width_in_m:
    values:
      - 2.2
  initializer:
    values:
      - glorot_uniform
      - glorot_normal
      - orthogonal
  is_periodic:
    values:
      - false
  learning_rate:
    values:
      - 0.0001
  n_epochs:
    values:
      - 20
  n_grad_steps_per_epoch:
    values:
      - 10000
  n_place_fields_per_cell:
    values:
      - 1
  optimizer:
    values:
      - sgd
      - adam
      - rmsprop
  place_cell_rf:
```

```
            values:
                - 0
        place_field_loss:
            values:
                - mse
        place_field_normalization:
            values:
                - none
        place_field_values:
            values:
                - cartesian
        readout_dropout:
            values:
                - 0
                - 0.5
        recurrent_dropout:
            values:
                - 0
                - 0.5
        rnn_type:
            values:
                - RNN
                - LSTM
                - UGRNN
                - GRU
        seed:
            values:
                - 0
                - 1
                - 2
        sequence_length:
            values:
                - 20
        surround_scale:
            values:
                - 1
        weight_decay:
            values:
                - 0
                - 0.0001
```

## F.2 Cartesian (Low Dimensional, Random)

```
method: random
metric:
    goal: minimize
    name: pos_decoding_err
parameters:
    activation:
        values: [
            'relu',
            'tanh',
            'sigmoid',
            'linear',
        ]
    batch_size:
        values: [30, 60, 90, 120, 150, 180]
    bin_side_in_m:
        values: [0.05]
```

```yaml
box_height_in_m:
  values: [2.2]
box_width_in_m:
  values: [2.2]
initializer:
  values: [
    'glorot_uniform',
    'glorot_normal',
    'orthogonal',
  ]
is_periodic:
  values: [False]
learning_rate:
  values: [0.005, 0.001, 0.0005, 0.0001]
n_epochs:
  values: [1, 4, 8, 12, 16]
n_grad_steps_per_epoch:
  values: [10000]
n_place_fields_per_cell:
  values: [ 1 ]
Ng:
  values: [1024]
Np:
  values: [2]
optimizer:
  values: [
    'adam',
    'rmsprop',
  ]
place_field_loss:
  values: [
    'mse',
  ]
place_field_values:
  values: [
    'cartesian',
  ]
place_field_normalization:
  values: [
    'none',
  ]
place_cell_rf:
  values: [
    0.
  ]
readout_dropout:
  values: [0., 0.05, 0.1, 0.2, 0.5 ]
recurrent_dropout:
  values: [0., 0.05, 0.1, 0.2, 0.5 ]
rnn_type:
  values: [
    'RNN',
    'LSTM',
    'UGRNN',
    'GRU',
  ]
seed:
  values: [ 0, 1, 2 ]
sequence_length:
```

```
        values: [20, 25, 30, 35, 40]
    surround_scale:
        values: [1.]
    weight_decay:
        values: [0., 0.0001, 0.0005, 0.001, 0.005, 0.01 ]
```

### F.3   Cartesian (High Dimensional)

```
method: grid
metric:
    goal: minimize
    name: pos_decoding_err
parameters:
    Ng:
        values:
            - 1024
    Np:
        values:
            - 512
    activation:
        values:
            - relu
            - tanh
            - sigmoid
            - linear
    batch_size:
        values:
            - 200
    bin_side_in_m:
        values:
            - 0.05
    box_height_in_m:
        values:
            - 2.2
    box_width_in_m:
        values:
            - 2.2
    initializer:
        values:
            - glorot_uniform
            - glorot_normal
            - orthogonal
    is_periodic:
        values:
            - false
    learning_rate:
        values:
            - 0.0001
    n_epochs:
        values:
            - 20
    n_grad_steps_per_epoch:
        values:
            - 1000
    n_place_fields_per_cell:
        values:
            - 1
    optimizer:
        values:
```

```
          - adam
    place_cell_rf:
      values:
        - 0
    place_field_loss:
      values:
        - mse
    place_field_normalization:
      values:
        - none
    place_field_values:
      values:
        - high_dim_cartesian
    readout_dropout:
      values:
        - 0
        - 0.5
    recurrent_dropout:
      values:
        - 0
        - 0.5
    rnn_type:
      values:
        - RNN
        - LSTM
        - UGRNN
        - GRU
    seed:
      values:
        - 0
        - 1
        - 2
    sequence_length:
      values:
        - 20
    surround_scale:
      values:
        - 0
    weight_decay:
      values:
        - 0
        - 0.0001
```

## F.4   Cartesian (High Dimensional, Random)

```
method: random
metric:
  goal: minimize
  name: pos_decoding_err
parameters:
  activation:
    values: [
      'relu',
      'tanh',
      'sigmoid',
      'linear',
    ]
  batch_size:
    values: [30, 60, 90, 120, 150, 180]
```

```
bin_side_in_m:
  values: [0.05]
box_height_in_m:
  values: [2.2]
box_width_in_m:
  values: [2.2]
initializer:
  values: [
    'glorot_uniform',
    'glorot_normal',
    'orthogonal',
  ]
is_periodic:
  values: [False]
learning_rate:
  values: [0.005, 0.001, 0.0005, 0.0001]
n_epochs:
  values: [1, 4, 8, 12, 16]
n_grad_steps_per_epoch:
  values: [1000]
n_place_fields_per_cell:
  values: [ 1 ]
Ng:
  values: [1024]
Np:
  values: [128, 256, 512]
optimizer:
  values: [
    'adam',
    'rmsprop',
  ]
place_field_loss:
  values: [
    'mse',
  ]
place_field_values:
  values: [
    'high_dim_cartesian'
  ]
place_field_normalization:
  values: [
    'none',
  ]
place_cell_rf:
  values: [
    0.0,
  ]
recurrent_dropout:
  values: [ 0., 0.05, 0.1, 0.2, 0.5  ]
readout_dropout:
  values: [ 0., 0.05, 0.1, 0.2, 0.5  ]
rnn_type:
  values: [
    'RNN',
    'LSTM',
    'UGRNN',
    'GRU',
  ]
seed:
```

```
      values: [ 0, 1, 2 ]
  sequence_length:
    values: [20, 25, 30, 35, 40]
  surround_scale:
    values: [
      0.
    ]
  weight_decay:
      values: [0., 0.0001, 0.0005, 0.001, 0.005, 0.01  ]
```

## F.5 Polar (High Dimensional)

```
method: grid
metric:
  goal: minimize
  name: pos_decoding_err
parameters:
  Ng:
    values:
      - 1024
  Np:
    values:
      - 512
  activation:
    values:
      - relu
      - tanh
      - linear
      - sigmoid
  batch_size:
    values:
      - 200
  bin_side_in_m:
    values:
      - 0.05
  box_height_in_m:
    values:
      - 2.2
  box_width_in_m:
    values:
      - 2.2
  initializer:
    values:
      - glorot_uniform
      - glorot_normal
      - orthogonal
  is_periodic:
    values:
      - false
  learning_rate:
    values:
      - 0.0001
  n_epochs:
    values:
      - 20
  n_grad_steps_per_epoch:
    values:
      - 1000
  n_place_fields_per_cell:
```

```
        values :
            − 1
    optimizer :
        values :
            − adam
    place_cell_rf :
        values :
            − 0
    place_field_loss :
        values :
            − mse
    place_field_normalization :
        values :
            − none
    place_field_values :
        values :
            − high_dim_polar
    readout_dropout :
        values :
            − 0
            − 0.5
    recurrent_dropout :
        values :
            − 0
            − 0.5
    rnn_type :
        values :
            − RNN
            − LSTM
            − UGRNN
            − GRU
    seed :
        values :
            − 0
            − 1
            − 2
    sequence_length :
        values :
            − 20
    surround_scale :
        values :
            − 0
    weight_decay :
        values :
            − 0
            − 0.0001
```

### F.6   Polar (High Dimensional, Random)

```
method : random
metric :
    goal : minimize
    name : pos_decoding_err
parameters :
    activation :
        values : [
            ' relu ' ,
            ' tanh ' ,
            ' linear ' ,
```

```
      'sigmoid',
   ]
batch_size:
   values: [30, 60, 90, 120, 150, 180]
bin_side_in_m:
   values: [0.05]
box_height_in_m:
   values: [2.2]
box_width_in_m:
   values: [2.2]
initializer:
   values: [
      'glorot_uniform',
      'glorot_normal',
      'orthogonal',
   ]
is_periodic:
   values: [False]
learning_rate:
   values: [0.005, 0.001, 0.0005, 0.0001]
n_epochs:
   values: [1, 4, 8, 12, 16]
n_grad_steps_per_epoch:
   values: [5000]
n_place_fields_per_cell:
   values: [ 1 ]
Ng:
   values: [1024]
Np:
   values: [128, 256, 512]
optimizer:
   values: [
      'adam',
      'rmsprop',
   ]
place_field_loss:
   values: [
      'mse',
   ]
place_field_values:
   values: [
      'high_dim_polar',
   ]
place_field_normalization:
   values: [
      'none',
   ]
place_cell_rf:
   values: [
      0.
   ]
readout_dropout:
   values: [0., 0.05, 0.1, 0.2, 0.5 ]
recurrent_dropout:
   values: [0., 0.05, 0.1, 0.2, 0.5 ]
rnn_type:
   values: [
        'RNN',
        'LSTM',
```

```
            'UGRNN' ,
            'GRU' ,
        ]
    seed :
        values : [ 0 , 1 , 2 ]
    sequence_length :
        values : [20 , 25 , 30 , 35 , 40]
    surround_scale :
        values : [0.]
    weight_decay :
        values : [0. , 0.0001 , 0.0005 , 0.001 , 0.005 , 0.01 ]
```

## F.7    Gaussian Place Cells

```
method : grid
metric :
    goal : minimize
    name : pos_decoding_err
parameters :
    Ng :
        values :
            − 1024
    Np :
        values :
            − 512
    activation :
        values :
            − linear
            − relu
            − tanh
            − sigmoid
    batch_size :
        values :
            − 200
    bin_side_in_m :
        values :
            − 0.05
    box_height_in_m :
        values :
            − 2.2
    box_width_in_m :
        values :
            − 2.2
    initializer :
        values :
            − glorot_uniform
    is_periodic :
        values :
            − false
    learning_rate :
        values :
            − 0.0001
    n_epochs :
        values :
            − 20
    n_grad_steps_per_epoch :
        values :
            − 10000
    n_place_fields_per_cell :
```

```
        values :
          − 1
  optimizer :
    values :
      − adam
  place_cell_rf :
    values :
      − 0.08
      − 0.1
      − 0.12
      − 0.14
      − 0.16
      − 0.2
      − 0.24
      − 0.28
  place_field_loss :
    values :
      − crossentropy
  place_field_normalization :
    values :
      − global
  place_field_values :
    values :
      − gaussian
  readout_dropout :
    values :
      − 0
      − 0.5
  recurrent_dropout :
    values :
      − 0
      − 0.5
  rnn_type :
    values :
      − RNN
      − LSTM
      − UGRNN
      − GRU
  seed :
    values :
      − 0
      − 1
  sequence_length :
    values :
      − 20
  surround_scale :
    values :
      − 1
  weight_decay :
    values :
      − 0.0001
```

## F.8   Gaussian Place Cells (Random)

```
method : random
metric :
  goal : minimize
  name : pos_decoding_err
parameters :
```

```
activation:
  values: [
      'linear',
      'relu',
      'tanh',
      'sigmoid',
  ]
batch_size:
  values: [30, 60, 90, 120, 150, 180]
bin_side_in_m:
  values: [0.05]
box_height_in_m:
  values: [2.2]
box_width_in_m:
  values: [2.2]
initializer:
  values: [
    'glorot_uniform',
    'glorot_normal',
    'orthogonal',
  ]
is_periodic:
  values: [False]
learning_rate:
  values: [0.005, 0.001, 0.0005, 0.0001]
n_epochs:
  values: [1, 4, 8, 12, 16]
n_grad_steps_per_epoch:
  values: [10000]
n_place_fields_per_cell:
  values: [
    1.0,
  ]
Ng:
  values: [1024]
Np:
  values: [128, 256, 512]
optimizer:
  values: [
    'adam',
    'rmsprop'
  ]
place_field_loss:
  values: [
    'crossentropy',
  ]
place_field_values:
  values: [
    'gaussian',
  ]
place_field_normalization:
  values: [
    'global',
  ]
place_cell_rf:
  values: [
    0.08,
    0.10,
    0.12,
```

```
            0.14,
            0.16,
            0.20,
            0.24,
            0.28,
            0.32,
            0.36,
            0.40,
        ]
    readout_dropout:
        values: [ 0., 0.05, 0.1, 0.2, 0.5   ]
    recurrent_dropout:
        values: [ 0., 0.05, 0.1, 0.2, 0.5   ]
    rnn_type:
        values: [
            'RNN',
            'LSTM',
            'UGRNN',
            'GRU',
        ]
    seed:
        values: [ 0, 1, 2]
    sequence_length:
        values: [20, 25, 30, 35, 40]
    surround_scale:
        values: [1.]
    weight_decay:
        values: [0., 0.0001, 0.0005, 0.001, 0.005, 0.01   ]
```

## F.9 Difference-of-Gaussians Place Cells

```
method: grid
metric:
    goal: minimize
    name: pos_decoding_err
parameters:
    Ng:
        values:
            - 1024
    Np:
        values:
            - 512
    activation:
        values:
            - relu
    batch_size:
        values:
            - 200
    bin_side_in_m:
        values:
            - 0.05
    box_height_in_m:
        values:
            - 2.2
    box_width_in_m:
        values:
            - 2.2
    initializer:
        values:
```

```
            - glorot_uniform
is_periodic:
  values:
    - false
learning_rate:
  values:
    - 0.0001
n_epochs:
  values:
    - 20
n_grad_steps_per_epoch:
  values:
    - 10000
n_place_fields_per_cell:
  values:
    - 1
optimizer:
  values:
    - adam
place_cell_rf:
  values:
    - 0.05
    - 0.1
    - 0.15
    - 0.2
    - 0.3
    - 0.4
    - 0.5
place_field_loss:
  values:
    - crossentropy
place_field_normalization:
  values:
    - global
place_field_values:
  values:
    - true_difference_of_gaussians
readout_dropout:
  values:
    - 0
    - 0.5
recurrent_dropout:
  values:
    - 0
    - 0.5
rnn_type:
  values:
    - RNN
    - LSTM
    - UGRNN
    - GRU
seed:
  values:
    - 0
    - 1
    - 2
sequence_length:
  values:
    - 20
```

```
surround_scale:
  values:
    - 1.5
    - 2
    - 2.5
    - 3
    - 4
    - 5
    - 6
weight_decay:
  values:
    - 0.0001
```

## F.10 Difference-of-Gaussians Place Cells (Random)

```
method: random
metric:
  goal: minimize
  name: pos_decoding_err
parameters:
  activation:
    values: [
      'linear',
      'relu',
      'tanh',
      'sigmoid',
    ]
  batch_size:
    values: [30, 60, 90, 120, 150, 180]
  bin_side_in_m:
    values: [0.05]
  box_height_in_m:
    values: [2.2]
  box_width_in_m:
    values: [2.2]
  initializer:
    values: [
      'glorot_uniform',
      'glorot_normal',
      'orthogonal',
    ]
  is_periodic:
    values: [False]
  learning_rate:
    values: [0.005, 0.001, 0.0005, 0.0001]
  n_epochs:
    values: [1, 4, 8, 12, 16]
  n_grad_steps_per_epoch:
    values: [10000]
  n_place_fields_per_cell:
    values: [ 1 ]
  Ng:
    values: [1024]
  Np:
    values: [128, 256, 512]
  optimizer:
    values: [
      'adam',
      'rmsprop'
```

```yaml
    ]
  place_field_loss:
    values: [
      'crossentropy',
    ]
  place_field_values:
    values: [
      'true_difference_of_gaussians'
    ]
  place_field_normalization:
    values: [
      'global',
    ]
  place_cell_rf:
    values: [
      0.08,
      0.10,
      0.12,
      0.14,
      0.16,
      0.20,
      0.24,
      0.28,
      0.32,
      0.36,
      0.40,
    ]
  recurrent_dropout:
    values: [ 0., 0.05, 0.1, 0.2, 0.5 ]
  readout_dropout:
    values: [ 0., 0.05, 0.1, 0.2, 0.5 ]
  rnn_type:
    values: [
      'RNN',
      'LSTM',
      'UGRNN',
      'GRU',
    ]
  seed:
    values: [ 0, 1, 2 ]
  sequence_length:
    values: [20, 25, 30, 35, 40]
  surround_scale:
    values: [
      1.5,
      2.,
      2.5,
      3.,
      3.5,
      4.,
      4.5,
      5.,
      5.5,
      6.,
    ]
  weight_decay:
    values: [ 0., 0.0001, 0.0005, 0.001, 0.005, 0.01 ]
```

## F.11 Difference-of-Softmaxes Place Cells

```
method: grid
metric:
  goal: minimize
  name: pos_decoding_err
parameters:
  Ng:
    values:
      - 1024
  Np:
    values:
      - 512
  activation:
    values:
      - relu
      - tanh
  batch_size:
    values:
      - 200
  bin_side_in_m:
    values:
      - 0.05
  box_height_in_m:
    values:
      - 2.2
  box_width_in_m:
    values:
      - 2.2
  initializer:
    values:
      - glorot_uniform
  is_periodic:
    values:
      - false
  learning_rate:
    values:
      - 0.0001
  n_epochs:
    values:
      - 20
  n_grad_steps_per_epoch:
    values:
      - 10000
  n_place_fields_per_cell:
    values:
      - 1
  optimizer:
    values:
      - adam
  place_cell_rf:
    values:
      - 0.08
      - 0.09
      - 0.1
      - 0.11
      - 0.12
      - 0.13
      - 0.14
```

```
            – 0.15
            – 0.16
            – 0.17
            – 0.18
            – 0.19
            – 0.2
            – 0.24
            – 0.28
            – 0.32
    place_field_loss:
        values:
            – crossentropy
    place_field_normalization:
        values:
            – global
    place_field_values:
        values:
            – difference_of_gaussians
    readout_dropout:
        values:
            – 0
            – 0.5
    recurrent_dropout:
        values:
            – 0
            – 0.5
    rnn_type:
        values:
            – RNN
            – LSTM
            – UGRNN
            – GRU
    seed:
        values:
            – 0
            – 1
            – 2
    sequence_length:
        values:
            – 20
    surround_scale:
        values:
            – 1.5
            – 2
            – 2.5
            – 3
            – 4
    weight_decay:
        values:
            – 0.0001
```

## F.12   Difference-of-Softmaxes Place Cells (Random)

```
method: random
metric:
    goal: minimize
    name: pos_decoding_err
parameters:
    activation:
```

```
    values: [
      'linear',
      'relu',
      'tanh',
      'sigmoid',
    ]
batch_size:
  values: [30, 60, 90, 120, 150, 180]
bin_side_in_m:
  values: [0.05]
box_height_in_m:
  values: [2.2]
box_width_in_m:
  values: [2.2]
initializer:
  values: [
    'glorot_uniform',
    'glorot_normal',
    'orthogonal',
  ]
is_periodic:
  values: [False]
learning_rate:
  values: [0.005, 0.001, 0.0005, 0.0001]
n_epochs:
  values: [20]
n_grad_steps_per_epoch:
  values: [10000]
n_place_fields_per_cell:
  values: [
    1.0,
  ]
Ng:
  values: [1024]
Np:
  values: [128, 256, 512]
optimizer:
  values: [
    'adam',
    'rmsprop',
  ]
place_field_loss:
  values: [
    'crossentropy',
  ]
place_field_values:
  values: [
    'difference_of_gaussians',
  ]
place_field_normalization:
  values: [
    'global',
  ]
place_cell_rf:
  values: [
    0.08,
    0.10,
    0.12,
    0.14,
```

```
          0.16 ,
          0.20 ,
          0.24 ,
          0.28 ,
          0.32 ,
          0.36 ,
          0.40 ,
          0.44 ,
          0.48 ,
      ]
  readout_dropout :
    values : [0. , 0.05 , 0.1 , 0.2 , 0.5]
  recurrent_dropout :
    values : [0. , 0.05 , 0.1 , 0.2 , 0.5]
  rnn_type :
    values : [
        'RNN' ,
        'LSTM' ,
        'UGRNN' ,
        'GRU' ,
    ]
  seed :
    values : [ 0 , 1 , 2 ]
  sequence_length :
    values : [20 , 25 , 30 , 35 , 40]
  surround_scale :
    values : [
        1.5 ,
        2. ,
        2.5 ,
        3.0 ,
        3.5 ,
        4.0 ,
        4.5 ,
        5.0 ,
        5.5 ,
        6.0 ,
    ]
  weight_decay :
      values : [0. , 0.0001 , 0.0005 , 0.001 , 0.005 , 0.01 ]
```

### F.13 Softmax-of-Differences Place Cells

```
method : grid
metric :
  goal : minimize
  name : pos_decoding_err
parameters :
  Ng :
    values :
      − 1024
  Np :
    values :
      − 512
  activation :
    values :
      − relu
  batch_size :
    values :
```

```
          - 200
bin_side_in_m:
  values:
    - 0.05
box_height_in_m:
  values:
    - 2.2
box_width_in_m:
  values:
    - 2.2
initializer:
  values:
    - glorot_uniform
is_periodic:
  values:
    - false
learning_rate:
  values:
    - 0.0001
n_epochs:
  values:
    - 20
n_grad_steps_per_epoch:
  values:
    - 10000
n_place_fields_per_cell:
  values:
    - 1
optimizer:
  values:
    - adam
place_cell_rf:
  values:
    - 0.09
    - 0.12
    - 0.15
    - 0.18
    - 0.21
    - 0.24
place_field_loss:
  values:
    - crossentropy
place_field_normalization:
  values:
    - global
place_field_values:
  values:
    - softmax_of_differences
readout_dropout:
  values:
    - 0
recurrent_dropout:
  values:
    - 0
rnn_type:
  values:
    - RNN
    - LSTM
    - UGRNN
```

```
            – GRU
seed :
    values :
        – 0
        – 1
        – 2
sequence_length :
    values :
        – 20
surround_scale :
    values :
        – 1.5
        – 2
        – 2.5
        – 3
weight_decay :
    values :
        – 0.0001
```

## F.14   Multiple Scales and Multiple Fields Difference-of-Softmaxes Place Cells

```
method : grid
metric :
    goal : minimize
    name : pos_decoding_err
parameters :
    Ng :
        values :
            – 1024
    Np :
        values :
            – 512
    activation :
        values :
            – relu
    batch_size :
        values :
            – 200
    bin_side_in_m :
        values :
            – 0.05
    box_height_in_m :
        values :
            – 2.2
    box_width_in_m :
        values :
            – 2.2
    initializer :
        values :
            – glorot_uniform
    is_periodic :
        values :
            – false
    learning_rate :
        values :
            – 0.0001
    n_epochs :
        values :
            – 20
```

```
n_grad_steps_per_epoch:
  values:
    - 10000
n_place_fields_per_cell:
  values:
    - Poisson( 2.0 )
    - Poisson( 3.0 )
optimizer:
  values:
    - adam
place_cell_rf:
  values:
    - 0.12
    - Uniform( 0.06 , 0.24 )
    - Uniform( 0.06 , 1.0 )
place_field_loss:
  values:
    - crossentropy
place_field_normalization:
  values:
    - global
place_field_values:
  values:
    - difference_of_gaussians
readout_dropout:
  values:
    - 0
recurrent_dropout:
  values:
    - 0
rnn_type:
  values:
    - RNN
    - UGRNN
seed:
  values:
    - 0
    - 1
    - 2
sequence_length:
  values:
    - 20
surround_scale:
  values:
    - 2
    - Uniform( 1.50 , 2.50 )
    - Uniform( 1.25 , 4.50 )
weight_decay:
  values:
    - 0.0001
```

### F.15   Nayebi et al. 2021 [47] Replication

```
method: grid
metric:
  goal: minimize
  name: pos_decoding_err
parameters:
  Ng:
```

```
      values:
        - 4096
Np:
   values:
     - 512
activation:
   values:
     - relu
batch_size:
   values:
     - 200
bin_side_in_m:
   values:
     - 0.05
box_height_in_m:
   values:
     - 2.2
box_width_in_m:
   values:
     - 2.2
initializer:
   values:
     - glorot_uniform
is_periodic:
   values:
     - false
learning_rate:
   values:
     - 0.0001
n_epochs:
   values:
     - 20
n_grad_steps_per_epoch:
   values:
     - 10000
n_place_fields_per_cell:
   values:
     - 1
optimizer:
   values:
     - adam
place_cell_rf:
   values:
     - 0.12
place_field_loss:
   values:
     - crossentropy
place_field_normalization:
   values:
     - global
place_field_values:
   values:
     - difference_of_gaussians
readout_dropout:
   values:
     - 0
recurrent_dropout:
   values:
     - 0
```

```
rnn_type:
  values:
    - RNN
    - LSTM
    - UGRNN
    - GRU
    - SRNN
seed:
  values:
    - 0
    - 1
    - 2
    - 3
    - 4
sequence_length:
  values:
    - 20
surround_scale:
  values:
    - 2
weight_decay:
  values:
    - 0.0001
```