# OpenReview forum: "No Free Lunch from Deep Learning in Neuroscience: A Case Study through Models of the Entorhinal-Hippocampal Circuit"
_NeurIPS.cc/2022/Conference — NeurIPS 2022 Accept_

### Official Review · Reviewer_PZh9 · 2022-07-08

**Rating:** 6
**Confidence:** 3
**Soundness:** 3 good
**Presentation:** 3 good
**Contribution:** 2 fair

**Summary:**

In this paper it is argued that the relevance of deep learning models for computational neuroscience should be carefully assessed by more explicitly stating the inductive biases needed to reproduce the target biological phenomena. Through extensive experiments, the authors show that recent deep learning simulations supposed to explain the emergence of grid cells in the hippocampal formation might have been driven by specific (and not always clearly justified) implementation choices, and use these findings to challenge the notion that deep learning is a panacea for building predictive models in neuroscience.

**Questions:**

Despite the results presented in this paper are interesting and provocative, I think that they do not undermine the key role that deep learning models are having for advancing computational (cognitive) neuroscience. The argument made by the authors should be better justified and contextualized in the broader field of computational cognitive neuroscience, not just on the specific case of entorhinal-hippocampal circuit: there are many successful stories of deep learning models that lead to important advances in neuroscience, and those should also be taken in consideration when drawing conclusions.

**Limitations:**

The authors did not explicitly mention any limitations of their research approach.

**Strengths And Weaknesses:**

Originality: I found this paper interesting and provocative. Related research is adequately referenced (actually, there might even be too many references and the authors might consider removing the less relevant ones). At the same time, the overall novelty of the work appears limited, since the basic argument is that neuroscientists exploring computational models are affected by strong confirmation biases and seek models that display the kind of emergent behavior they are looking for. Though important, such claim is nothing new in the modeling literature.

Quality: The authors put lot of efforts to reproduce previously published models (through open access source code) and to carefully investigate the impact of hyperparameters in driving model behavior. Still, the hyperparameter sweeps were not exhaustive, and it seems that in some cases the level of detail required to reproduce the original simulations was not achieved. Though the latter issue is mostly due to the limited information provided in the original publications, it might still have affected the present findings. Moreover, it seems that in some cases the results and interpretations depend on the arbitrary choice of some threshold (e.g., grid scores in Fig. 2) over which there might not be an objective consensus in the literature.

Clarity: The paper is well-written and appropriately structured. The reference list should be double-checked and shortened (there are also some duplicates, e.g., [20] vs. [21]). Readability of the figures can be improved, for example by better organizing the spacing layout and by increasing font size.

Significance: This paper could stimulate further research work. However, in my opinion the fact that a result (e.g., emergence of lattice cells) depends on the careful choice of some model’s hyperparameters does not necessarily undermine the relevance of the model. Rather, it calls for further investigations to better establish how and why certain hyperparameters could assume such specific values. The work carried out by the authors is certainly useful (and remarkable): given that the replication crisis is extending from experimental sciences to modeling sciences, pointing out to issues with published models is a good way to better define their implicit assumptions and possible shortcomings. However, I am not particularly impressed by the finding that only certain combinations of hyperparameters lead to the simulation of relevant emergent phenomena. For example, I do not think that deep learning models should necessarily be robust to variations in the type of activation function: if only one kind of function “works”, then the modeler task should be to try explaining which biological features might justify the presence of such activation function. Overall, these findings/discussions seem more appropriate for a computational neuroscience conference / journal (and maybe one focused on hippocampus).

---

> ### Author Response · Authors · 2022-08-02
> **Response to PZh9 (2/2)**
>
> > Overall, these findings/discussions seem more appropriate for a computational neuroscience conference / journal (and maybe one focused on hippocampus).
>
> The goal of our paper is to study and communicate the potential risks and rewards of using deep neural networks to model neural systems of interest. Our hope is that our findings and our message contribute insights to anyone interested in the intersection of deep learning, cognitive science and neuroscience.
>
> The NeurIPS call for papers explicitly invites a paper like ours, which falls under the invited NeurIPS topics of (1) "Neuroscience and Cognitive Science," (2) "Machine Learning for Sciences", and (3) "Deep Learning." Even if our work doesn't fit neatly into one of these categories, NeurIPS additionally states, "we welcome interdisciplinary submissions that do not fit neatly into existing categories."
>
> Moreover, many of the key preceding papers we build on were published at NeurIPS (Sorscher et al. 2019, Nayebi et al. 2021) and ICLR (Cueva & Wei 2018). As a matter of principle, evaluation of papers should be made accessible to the same audiences that read the earlier papers, by being published in the same/similar venues. If venues accepted papers, but not possible concerns with those papers, then science would suffer from claims that are possibly misleading, insufficiently substantiated or altogether unreproducible.
>
>
> > the results and interpretations depend on the arbitrary choice of some threshold (e.g., grid scores in Fig. 2) over which there might not be an objective consensus in the literature
>
> We are similarly concerned, but our paper is not the source of this subjectivity; this subjectivity already exists in the literature. Moreover, the grid score metric is itself subject to much debate and different experimental research groups compute it differently or use an alternative metric. That said, we display our results without thresholds (Figures 2b, 4b, 5b) or with multiple different thresholds (Figure 4a), showing that our results qualitatively hold regardless of what threshold is applied.
>
> Thank you for your keen eye! That is correct and we removed the duplicate.
>
> > the authors might consider removing the less relevant [citations]
>
> Following your suggestion, we have pared down the number of citations. Do let us know if you feel other citations should be removed.
>
> > there are also some [duplicates citations] increasing font size
>
> Again, thank you for this constructive feedback! We worked to increase the font size across all figures; if any figure font size is still too small, please let us know.

---

> > ### Comment · Reviewer_PZh9 · 2022-08-04
> > **A more balanced manuscript**
> >
> > I thank the Authors for their thoughtful responses. I read the updated version of the manuscript, which indeed appears clearer and more balanced. Readability of the figures also improved.
> >
> > Regarding the hyperparameters sweeps reported in Appendix E, I am still wondering whether some hyperparameters (e.g., batch_size, learning_rate, n_epochs, dropout, weight_decay) should be sampled from a wider range, since they might have a significant impact on the learning convergence and/or the final weights of the network. Maybe a random search rather than a grid search would help in better exploring the optimal configurations without increasing too much the computational cost?
> > Regarding the reproducibility of original simulations, I was referring to Section 9 (now Section 7), which raises an interesting issue about using linear regression to perform comparisons between ANNs and neural data. Have the Authors tried to reach out the authors of the target NeurIPS spotlight to ask for clarifications / analysis code?
> >
> > Overall, I agree that this might be the first paper challenging the “grid-cells DL success story”, which could be of interest to some of the NeurIPS attendees. What I find less original, though, is the main argument that “neuroscientists exploring computational models are affected by strong confirmation biases and seek models that display the kind of emergent behavior they are looking for”, since this is something that is well-known in the broader modeling community.
> >
> > Considering all this, I increased my score from 3 to 5.

---

> > > ### Author Response · Authors · 2022-08-05
> > > **Response to Reviewer PZh9**
> > >
> > > Thank you for getting back to us so quickly! We’re glad to hear you found the updated version of the manuscript clearer and more balanced, thanks to the comments of our reviewers.
> > >
> > > > I am still wondering whether some hyperparameters (e.g., batch_size, learning_rate, n_epochs, dropout, weight_decay) should be sampled from a wider range, since they might have a significant impact on the learning convergence and/or the final weights of the network
> > >
> > > Thank you for clarifying your concerns with our sweeps (and for reading our sweeps in detail - we very much appreciate that level of attention). We focused on those hyperparameters because previous publications found they worked best for creating grid cells and we wanted our results to maximally favor their emergence.
> > >
> > > That said, we fully agree we could broaden our sweeps, and we will use random search to do so. We will update the paper and notify you once results are available.
> > >
> > > > Have the Authors tried to reach out to the authors of the target NeurIPS spotlight to ask for clarifications / analysis code?
> > >
> > > All models in Figure 7 have been trained from the code repository of the NeurIPS spotlight paper (Nayebi et al. 2021), who gave us private access to it, using their hyperparameters. However, the analysis code is not open source and the neural data is not public; we have asked on several occasions for access to the neural data, but we did not receive it. Thus, our conclusions in Section 7 are partly speculative, and we have edited the language in the section to more clearly reflect this fact. We intend for this section to be a call to the community to more closely investigate the techniques and results of comparisons between artificial and biological networks, in particular how the dimensionality of representations affects linear regression scores.
> > >
> > > Our speculative prediction, that high dimensional ANNs may outperform low dimensional models in matching neural data more simply because of dimensionality rather than because of a detailed similarity, seems to find very recent support in the context of vision models, see the work of Elmoznino & Bonner 2022, https://www.biorxiv.org/content/10.1101/2022.07.13.499969v1: Fig 3a.
> > >
> > > > What I find less original, though, is the main argument that “neuroscientists exploring computational models are affected by strong confirmation biases and seek models that display the kind of emergent behavior they are looking for” since this is something that is well-known in the broader modeling community.
> > >
> > > Here we respectfully disagree, because this issue is a serious and unappreciated or underappreciated problem for Neuroscience at large; it is the whole reason for why we felt compelled to write this paper! Neuroscientists are not usually experts in deep learning, and are typically biologists by training, who take a relatively at-face-value view of results from ANNs. Training ANNs and demonstrating some resemblance to neural data is a widespread enterprise in Neuroscience now, but the field lacks a critical diagnosis of what works, what is predictive, what is post-hoc, and what are the right messages/outcomes to be taken from the models. The extremely high-profile (Nature paper, Banino et al.; Neurips spotlight, Nayebi et al., and others) and specific claim made by grid cell ANN model papers is that path integration leads to grid cells, which we show here to not be true; yet these works have been used (and continue to be used) as some of the most influential examples in Neuroscience of how deep learning sheds light on neural representation. There is not a general understanding in the community that these results are post-hoc, even if that appreciation is present in the DL field given the extensive experience of DL practitioners with ANN models.  We are showing that had grid cells not been known to exist a priori, the models would not have found them, and hence the claims of being predictive or of having identified the correct optimization function are both inaccurate.

---

> > > ### Author Response · Authors · 2022-08-09
> > > **Response to Reviewer PZh9**
> > >
> > > Hi Reviewer PZh9,
> > >
> > > We want to follow up with you regarding our last message, specifically on 2 fronts.
> > >
> > > 1) You recommended wider hyperparameter sweeps, and suggested perhaps using random search rather than grid search. We took your advice to heart and ran significantly wider random sweeps (details added to Appendix). Doing so increased our number of runs from ~500 (in the first draft) to ~5,000 (in the previous draft) to ~11,000 (in the newest draft, ~~which will be posted in <2 hours~~). Qualitatively, our results did not change. We did find two small differences that don't affect the main scientific contributions:
> > >
> > > 1a) A slightly higher percentage of runs failed to achieve low position decoding error. This seems reasonable since some hyperparameter combinations might possibly make optimization harder (e.g. weight regularization too high, learning rate too large, longer trajectories causing exploding/vanishing gradients, etc.)
> > >
> > > 1b) While the *distribution* of grid scores for high-dimensional polar encodings (akin to speed & head direction cells) did not change (Figure 2a, top), more runs (~20% of N=1171 runs) did achieve a higher *max* grid score (Figure 2a, bottom). We investigated and found that these runs learnt a small number (typically <5 units of 1024 total units) of units with triangular-like representations that achieve a high grid score but are *not* grid cells (Figure 2c, Polar). This can be explained because the grid score doesn't actually score lattices, but rather  scores 60 degree rotational symmetry.
> > >
> > > 2) We are curious to know if you have any additional thoughts on our previous message, specifically our response to the notion that “neuroscientists exploring computational models are affected by strong confirmation biases and seek models that display the kind of emergent behavior they are looking for” is well-known in the broader modeling community.
> > >
> > > Here we respectfully disagree, because this issue is a serious and unappreciated or underappreciated problem for Neuroscience at large; it is the whole reason for why we felt compelled to write this paper! Neuroscientists are not usually experts in deep learning, and are typically biologists by training, who take a relatively at-face-value view of results from ANNs. Training ANNs and demonstrating some resemblance to neural data is a widespread enterprise in Neuroscience now, but the field lacks a critical diagnosis of what works, what is predictive, what is post-hoc, and what are the right messages/outcomes to be taken from the models. The extremely high-profile (Nature, Banino et al.; NeurIPS Spotlight, Nayebi et al., and others) and specific claim made by grid cell ANN model papers is that path integration leads to grid cells, which we show here to not be true; yet these works have been used (and continue to be used) as some of the most influential examples in Neuroscience of how deep learning sheds light on neural representation. There is not a general understanding in the community that these results are post-hoc, even if that appreciation is present in the DL field given the extensive experience of DL practitioners with ANN models. We are showing that had grid cells not been known to exist a priori, the models would not have found them, and hence the claims of being predictive or of having identified the correct optimization function are both inaccurate.
> > >
> > >
> > > Edit: We just learned we can no longer post a revised manuscript. We do promise that we ran your sweeps and that Figure 2 has been updated accordingly.

---

> > > > ### Comment · Reviewer_PZh9 · 2022-08-09
> > > > **A stronger paper**
> > > >
> > > > I am glad to see that the authors spent extra efforts to improve the hyperparameter search. Considering the appreciation demonstrated by the other Reviewers, I feel I might have been a bit biased in my initial (negative) judgement, probably due to my background in both neurocognitive modeling and deep learning. I am thus willing to further raise my score to 6.

---

> ### Author Response · Authors · 2022-08-02
> **Response to PZh9 (1/2)**
>
> Thank you for your detailed review of our paper! We were thrilled to hear that you found our results “interesting and provocative” and “useful (and remarkable).”
>
> In response to your thoughtful criticisms and those of the other reviewers, we have rewritten significant portions of our paper, so we ask that you please take a fresh look. In particular, we rewrote or revised the Discussion, Results (section on match to neural data), Introduction, and (to a lesser extent) Abstract, revamped many of our figures to improve clarity and added many additional sections to our Appendix. We also increased the number of trained networks from ~500 to ~5000.
>
> In response to your comments:
>
> > Though important, such claim is nothing new in the modeling literature.
>
> The claim that grid cells emerge from deep networks trained to path integrate is one of the great success stories of deep learning in neuroscience that has been published in top-tier machine learning conferences (NeurIPS, ICLR) and scientific journals (Nature, Cell). To the best of our knowledge, our paper is the first to challenge this success story. While others might have made thematically similar claims in other (sub)fields, our paper is novel in questioning path integrating deep network models of grid cells.
>
> > the hyperparameter sweeps were not exhaustive
>
> Could you please clarify why you find our hyperparameter sweeps insufficiently exhaustive? If you have specific experiments you would like us to run, we would be happy to do so. Since submitting this paper, we have updated our results with significantly more networks (previously ~500, now ~5000).
>
> > it seems that in some cases the level of detail required to reproduce the original simulations was not achieved
>
> We are fully able to reproduce the results of previous papers using the authors’ publicly available code (Figures 2 through 5 all contain data reproduced from previous papers’ approaches). Could you please clarify what exactly you are referring to, and what results you would like to see?
>
> > However, in my opinion the fact that a result (e.g., emergence of lattice cells) depends on the careful choice of some model’s hyperparameters does not necessarily undermine the relevance of the model.
>
> The hyperparameters that produce grid units (single-field single-scale uniformly-distributed place cells with isotropic difference-of-Softmax tuning curves) are known to be inconsistent with multiple properties of biological place cells. Biological place cells are not single field, not single scale, not uniformly distributed, not isotropic, and do not display difference-of-Softmax tuning curves. In fact, experimentally measured place cells are at the opposite extreme e.g. they are heterogeneous with multiple scales in several species (Eliav et al. Science 2021 for bats, Rich et al. Science 2014 for rodents), they do not uniformly cover space tending to cluster around boundaries and landmarks. We also show in our paper that the path integrating networks learn grid units that are inconsistent with biological grid cells (e.g., networks learn only a single grid module). For these reasons, we think that it is less likely that hyperparameters could assume such values biologically. We worked to make this point more clear across our Background, Results and Discussion sections.
>
> > This paper could stimulate further research work. [...] it calls for further investigations to better establish how and why certain hyperparameters could assume such specific values.
>
> This point, that our paper calls for further investigation, is a reason to support its publication. By publishing this work, other researchers can take note of this apparent gap between properties of biological place cells and the necessary properties of artificial place cells, and can investigate whether there is some way to reconcile the apparent gap.

---

### Official Review · Reviewer_B3Mb · 2022-07-09

**Rating:** 6
**Confidence:** 4
**Soundness:** 3 good
**Presentation:** 2 fair
**Contribution:** 3 good

**Summary:**

The theme of this paper is to demonstrate that directly using deep learning models as models of brain functions might lead to wrong or incomplete conclusions. By performing extensive hyper-parameter searches, authors show that the emergence of grid cell patterns by training a path integration RNN depends solely on particular design choices such as DoG firing fields rather than the path integration task itself. Such observations can hardly support conclusions made by previous papers that the emergence of grid cell patterns is due to performing the path integration task by an RNN. As a result, authors argue that neuroscientists should be more cautious to use deep learning models directly as models of brain function.

**Questions:**

1. Whether there are other examples in neuroscience that use deep learning but lead to misleading conclusions?
2. Any new ideas to correctly and cautiously use deep learning models to interpret brain functions?
3. Possible comparisons of expert-designed computational models (CANN) and deep learning models?

**Limitations:**

Yes.

**Strengths And Weaknesses:**

Strengths:
+ The considered question is important and remains an open and challenging one.
+ The perspective adopted by this paper is interesting, namely to demonstrate fundamental limitations of previous approaches.

Weakness:
- This paper does not figure out any potential solutions to mitigate the observed mismatch between deep learning models and brain functions.
- Many assumptions such as DoG firing fields or special regularizations are indeed clearly claimed in previous papers. So it seems natural that without these assumptions the grid cell patterns should disappear.
- Only one example (grid cells) is used to show the no-free-lunch claim.

---

> ### Author Response · Authors · 2022-08-02
> **Response to B3Mb (2/2)**
>
> > Only one example (grid cells) is used to show the no-free-lunch claim.” and “Whether there are other examples in neuroscience that use deep learning but lead to misleading conclusions?
>
> The preceding deep learning grid cells papers are numerous and span multiple top-tier scientific journals (Nature, Cell) and machine learning conferences (NeurIPS, ICLR). We felt it was important to deeply and carefully analyze one body of work - in which deep learning has been claimed as a major neuroscience modeling success story - as opposed to cursorily covering multiple bodies of work. We also sought to exhaustively illustrate different analyses and sweeps that are appropriate for studying the robustness of previously published claims. We believe our paper could serve as a template to ask similar questions and perform detailed analyses for other areas/brain systems. To show such careful analyses for another brain system with as much rigor would be beyond the scope of a single paper.

---

> > ### Comment · Reviewer_B3Mb · 2022-08-08
> > **Response to author rebuttal**
> >
> > Thank you to authors for their responses. I read the new version of this paper. It has been improved a lot and addressed my concerns. Considering this, I raise my rating from 4 to 6. In addition, I have one further question, could you succinctly discuss the relative merits and disadvantages of CANN models and deep learning models for grid cell modeling?

---

> > > ### Author Response · Authors · 2022-08-09
> > > **Response to Reviewer B3Mb**
> > >
> > > We're very happy to hear you feel our paper was improved by yours and other reviewers' feedback.
> > >
> > > > could you succinctly discuss the relative merits and disadvantages of CANN models and deep learning models for grid cell modeling?
> > >
> > > We'd be delighted to. We have added a paragraph to the Discussion section of the newest draft, ~~which will be posted in <2 hours~~.
> > >
> > > Edit: We just found that it is now too late to post a revised manuscript.

---

> ### Author Response · Authors · 2022-08-02
> **Response to B3Mb (1/2)**
>
> Thank you for your detailed review! We are grateful to hear you agree that “Such [empirical] observations can hardly support conclusions made by previous papers that the emergence of grid cell patterns is due to performing the path integration task by an RNN.” This is the primary message we strive to convey with this paper, and we feel it is critical for the field to hear this message. We have toned down our language to reflect this in the main text of the paper. For other neural systems, we hope that our grid cell findings are an important cautionary tale and a template for future possible investigations.
>
> In response to your thoughtful criticisms and those of the other reviewers, we have rewritten significant portions of our paper, so we ask that you please take a fresh look. In particular, we rewrote or revised the Discussion, Results (section on match to neural data), Introduction, and (to a lesser extent) Abstract, revamped many of our figures to improve clarity and added many additional sections to our Appendix. We also increased the number of trained networks from ~500 to ~5000.
>
> Below we directly address some of the weaknesses you identified.
>
> > This paper does not figure out any potential solutions to mitigate the observed mismatch between deep learning models and brain functions.
>
> On the specific topic of grid cells, we extended our Discussion section with a forward-looking and detailed alternative optimization approach for obtaining grid cells.
>
> On the more general question of mismatch between deep learning models and brains, this is a difficult question and our main message here is that the community must think hard about how to bridge the gap. This is why we believe it is important to publish our work: our work highlights to the community what areas are in dire need of additional work to better understand and mitigate said mismatch.
>
> We have added possible suggestions to our Discussion section. We believe that biological knowledge/constraints are important, should be acknowledged, and these are important to explore: under what conditions a particular feature does or does not emerge is the actual scientific question. We also suggest that, in the absence of studying detailed biological constraints and their effects of tuning, manifold-/population-level conclusions might be safer to draw, if these illustrate what is required to solve the task, rather than single-neuron curves. We also believe causal experiments using edge cases (e.g. adversarial examples in vision, metamers in audition) are also good techniques for testing how well deep learning models generalize in predicting the brain by leveraging test data far away from the “typical” data distribution.
>
> > Many assumptions such as DoG firing fields or special regularizations are indeed clearly claimed in previous papers.
>
> Unfortunately, these assumptions are not clearly claimed in previous papers, and they have led to endemic misunderstanding and confusion in the field. Most neuroscientists, who may not be experts in the subtleties and details of hyperparameter tuning and deep learning in general, believe that the simple objective of path integration is indeed sufficient for the emergence of grid cells, on the basis of the papers we are examining in this work. For instance, Banino et al. 2018 wrote “Notably, therefore, our results show that grid-like representations reminiscent of those found in the mammalian entorhinal cortex emerge in a generic network trained to path integrate,” and  Sorscher et al. 2019 wrote “Why do these diverse architectures, across diverse tasks (both navigation and autoencoding), all converge to a grid-like solution?” when in fact most don’t. Additionally, the first row of Sorscher et al.’s Figure 1 shows Gaussian firing fields creating lattices, despite the fact that Sorscher et al.’s analytical results, our empirical sweeps and our analytical results all suggest that such networks should not exist (simple Gaussian readouts should not produce grid cells). Lastly, as best as we can tell, Nayebi et al. 2021 did not mention many important assumptions, e.g., using only DoS firing fields. In sum, these assumptions you mention may indeed be available, but only to the highly knowledgeable subject expert willing to dig deep into these details of the code repositories.
>
> Our paper demonstrates the extent to which previous results are brittle. The original papers did not present any investigations like the ones we have in our paper.

---

### Official Review · Reviewer_M9Ch · 2022-07-10

**Rating:** 7
**Confidence:** 2
**Soundness:** 3 good
**Presentation:** 3 good
**Contribution:** 4 excellent

**Summary:**

This paper casts doubt on whether DNNs can serve as good mechanistic models to understand the brain. In particular, this paper revisits the existing literature on how task-related loss functions (e.g., path integration) of RNNs produce grid-like neural tuning, a key finding supporting the applications of DL models in neuroscience. The authors leveraged the published code in earlier work and performed large-scale hyperparameter sweeps across several dimensions, including network architectures, activation functions, optimizers, supervised targets, loss functions, and other miscellaneous aspects of model training. Surprisingly, almost all trained networks learn to perform path integration tasks very well but only a few exhibit grid-like codes. In particular, they found that networks whose neurons are assumed to have DoG receptive fields exhibit grid-like codes, but even under this regime grid-like cells only emerge for a small range of receptive fields parameter values. The authors also provide theoretical explanations from the perspective of Fourier analysis for why only a small range of parameter values of DoG tuning curve can lead to lattice cells. The authors also criticize for inconsistency between method descriptions in paper and true code implementations. Lastly, the authors investigate the intrinsic dimensionalities of RNN and how it could help predict neural activity.

**Questions:**


* Choosing appropriate hyperparameters is a fundamental aspect of machine learning and neuroscience research. For example, in the majority of instances, we say a model "state-of-the-art" only on the condition of certain hyper-parameters. Commonly, a model may no longer be the best for other hyper-parameter settings. In neuroscience, certain neural properties only hold for some task parameters (e.g., stimulus size, contrast). This is why reproducibility rises as an issue in neuroscience because readers typically have no idea of the boundary of appropriate hyper-parameters/experimental settings. In my opinion, hyper-parameter _per se_ is not an issue. The issue here is whether a paper is transparent for the hyper-parameters being chosen such that others can reproduce their results. Do you think a claim can be accepted only after it is confirmed on every possible hyper-parameters? I don't think so. I hope the authors could clarify this.
* In section 9. The authors investigated the correlation between dimensionalities and neural prediction scores, and speculate that DL models outperform alternatives because they provide higher-dimensional bases. I don't quite get the logic. We can evaluate a model from many perspectives. One important aspect is its explanatory power for data. Higher-dimensional bases usually indicate richer representational power albeit possible suffering of overfitting. I don't think high-dimensional bases _per se_ is bad if it is well cross-validated. Also, this cannot be easily generalized to vision and language models. Vision DNN models typically have no temporal domain and they are not dynamic systems.
* The authors seem to overly generalize their pessimism to other fields in neuroscience, e.g., vision and language. This is another form of exaggeration in the opposite direction. I do agree that this paper reports a disaster failure of RNN models in explaining grid-like codes. But critics should be reviewed case by case. In other words, unless similar analyses are done in vision or language models to illustrate their deficiency, we should not make such analogy to DNN in other fields.
* One central claim in the Discussion part is universally true and applies to all computational models, not specific to DNNs. Given non-linearity in the majority of computational models, there always exists the possibility of multiple solutions for a given loss and multiple losses for a given solution. The question is the extent of such bad interpretability. For example, trained sparse coding models contain neurons whose receptive fields bear striking resemblance to receptive fields in V1. And people conclude that sparse coding may be one of the loss functions the visual system attempts to optimize. We can certainly find other loss functions giving rise to similar results. But the sparse coding studies are still informative and valuable.

**Limitations:**

I find no specific limitations here.

**Strengths And Weaknesses:**

Before going through this paper, I want to be transparent that I am familiar with DNN and RNN training but not an active researcher in the field of gird cells and spatial navigation. So I may misunderstand some aspects of this paper. I explicitly list the parts I don't understand here for meta-reviewers or area chairs to accurately evaluate this manuscript.

I understand the majority of this paper except that:

* The  Fourier analysis part for why some specific DoG tuning curve values can lead to grid-like codes in Section 6.
* Related to the above point, the difference between DoG and DoS in Section 8.



# Strengths

* This paper is interesting and should be published. All researchers in the field should be aware of this. I appreciate the thorough analyses on where the trained networks do or do not produce grid-like codes. Although DNNs have recently been touted as good models in neuroscience, the field needs this sober consideration and careful examination of the validity of existing results. People should be more objective and critical to papers even published in prestigious journals.
* The analyses are good. At least I cannot find any fundamental flaw in terms of analyses.
* In Discussion, the authors also sketch out the possible ingredients that could promote existing RNNs to obtain grid-like codes, which is very constructive.


# Weakness:

My major concern is that, although we should criticize the seemingly exaggerated claims in existing literature, we should also be careful and objective to the claims in this paper. I think some claims here are going to the opposite extreme. See my questions below.

Minor
* Ref. 20 and 21 seem identical

---

> ### Author Response · Authors · 2022-08-02
> **Response to M9Ch (1/1)**
>
> Thank you for your detailed review! We are delighted that you agree that our work reports a possible “failure of RNN models in explaining grid-like codes” and grateful to hear you say that “researchers in the field should be aware of this.” On a personal note, we agree wholeheartedly with your statement that “the field needs this sober consideration and careful examination of the validity of existing results. People should be more objective and critical to papers even published in prestigious journals.”
>
> In response to your thoughtful criticisms and those of the other reviewers, we have rewritten significant portions of our paper, so we ask that you please take a fresh look. In particular, we rewrote or revised the Discussion, Results (section on match to neural data), Introduction, and (to a lesser extent) Abstract, revamped many of our figures to improve clarity and added many additional sections to our Appendix. We also increased the number of trained networks from ~500 to ~5000.
>
> > My major concern is that, although we should criticize the seemingly exaggerated claims in existing literature, we should also be careful and objective to the claims in this paper. I think some claims here are going to the opposite extreme.
>
> We agree. We rewrote our paper accordingly, taking this guidance into account. We made certain to tone down our language, especially in reference to other areas of neuroscience.
>
> > The issue here is whether a paper is transparent for the hyper-parameters being chosen such that others can reproduce their results. Do you think a claim can be accepted only after it is confirmed on every possible hyper-parameters? I don't think so. I hope the authors could clarify this.
>
> Again, we agree wholeheartedly that transparency and honest evaluation of hyperparameters is of paramount importance. We do not believe that a claim can only be accepted if confirmed on all possible hyperparameters, and have clarified this in our discussion.
>
> > In section 9 [...] I don't quite get the logic. We can evaluate a model from many perspectives. One important aspect is its explanatory power for data. Higher-dimensional bases usually indicate richer representational power albeit possible suffering of overfitting. I don't think high-dimensional bases per se is bad if it is well cross-validated.” and ““Also, this cannot be easily generalized to vision and language models. Vision DNN models typically have no temporal domain and they are not dynamic systems.””
>
> We have rewritten the text corresponding to this section to be more speculative and shrunk this section. In particular we have added: “We caution that this correlation between rate map dimensionality and neural predictivity is not strong evidence“. Our prediction for vision very recently received supporting evidence on BioRxiv (Elmoznino & Bonner, https://www.biorxiv.org/content/10.1101/2022.07.13.499969v1: Fig 3a); that said, we agree this hypothesis is very preliminary and rewrote the section accordingly.
>
> Cross validation, while incredibly useful, is not a panacea and can yield misleading results (see this wonderful preprint by Kenneth Harris https://www.biorxiv.org/content/10.1101/2020.11.29.402719v3). We’ve removed this discussion from our manuscript, but we suspect something similar could happen and needs to be investigated more rigorously.
>
> > Ref. 20 and 21 seem identical
>
> Thank you for your keen eye! That is correct and we removed the duplicate citation.
>
> > I understand the majority of this paper except that: The Fourier analysis part for why some specific DoG tuning curve values can lead to grid-like codes in Section 6.
>
> We improved our exposition of the Fourier analysis in the paper, which hopefully will make the topic more approachable, and added a section to the Appendix with additional details for any curious readers. That said, the main takeaway from this section is that our empirical findings can be predicted/explained by earlier non-deep learning, first-principles modeling papers such as Burak & Fiete 2009 or Dordek, Soudry, Meir & Derdikman 2016. We make this point to question whether the deep learning papers we study have contributed beyond (or even up to) what was previously known from non-deep learning models.

---

> > ### Comment · Reviewer_M9Ch · 2022-08-09
> > **a substantially improved paper**
> >
> > Dear author,
> > I read your responses and appreciate your efforts in revising the paper. I've no issues in terms of your responses. I am positive to your paper from the very beginning.
> >
> > I have the last request. I think the Fourie analysis part should be highlighted more as it is a theoretical proof of why some existing results should never happen. A theoretical interpretation is critical here as it could be further confirmed or falsified by readers.
> >
> > Good work.

---

### Official Review · Reviewer_5qo3 · 2022-07-11

**Rating:** 8
**Confidence:** 2
**Soundness:** 4 excellent
**Presentation:** 4 excellent
**Contribution:** 3 good

**Summary:**

This work critically reanalyzes previously published deep neural network models of grid cell emergence. It shows that grid-cell-like representations arise only given overly specific implementations of the path integration task and might even depend on undocumented implementation choices. Hence, the training task alone does not predict the emergence of grid cells. The authors suggest that the overly specific occurrence of grid-cell-like representations in NNs trained on path integration reflects the posthoc nature of these modeling studies: Presumably, the implementations of these models were informally tweaked until they reproduced the existing biological phenomenology. The authors analyze the emergent grid-cell-like representations and provide some analytical results on why they arise. Finally, the authors argue that neural network-based modeling must rely more strongly on biological and theoretical constraints. Multiple training objectives may map to the same neural activity pattern. Conversely, the "correct" objective will not lead to the empirical neural activity pattern without the relevant biological constraints.

**Questions:**

* Why shouldn't we think about the specificity of grid-cell-like representation emergence as a feature rather than a bug of deep neural network modeling? (see above).
* How was the grid score defined? I could not find the relevant equation.
* Is there a repository with the code required for reproducing the study's experiments and results? Adhering to open research practices would increase the impact of this work (and my assigned NeurIPS score). It would also be fairer, given that this study was enabled by the open research practices of the criticized works.

comments:
* Pie charts (figure 1) are an inefficient, outdated, and bias-prone visualization method. An empirical CDF or histogram of grid scores or a scatter plot of position error vs. grid score would convey much more information to the readers.

**Limitations:**

Limitations seem to be sufficiently addressed.

**Strengths And Weaknesses:**

This is a deep, quantitive critique of current connectionist approaches to explaining grid cells. It is systematic and well presented.

Having said that, I am left unconvinced by the claim of many-to-one mappings of training tasks and activity patterns (figure 8 top); the empirical results seem to demonstrate the opposite pattern - surprisingly high sensitivity of grid-cell emergence to the specific operationalization of the training objective. In principle, this result can be interpreted as a strength of the NN-based modeling approach - we can reject incorrect operationalization. This sensitivity contrasts with traditional neurobiological "word models" that always magically work since they do not commit to an implementation. Therefore, I believe it would be fairer and more constructive to frame the paper's results as a description of a considerable remaining modeling gap and avoid sweeping statements such as  "...that one often gets neither".

---

> ### Author Response · Authors · 2022-08-02
> **Respones to 5qo3 (2/2)**
>
> > How was the grid score defined?
>
> At a high level, the grid score measures how correlated a rate map is with itself after being rotated by particular angles. The 60 degree grid score is specifically given by
>
> (corr[60] + corr[120]) / 2 - (corr[30] + corr[90] + corr[150]) / 3
>
> To ensure we didn’t make a mistake in our implementation, we used the same grid scorer used by both Banino et al. 2018 (https://github.com/deepmind/grid-cells/blob/master/scores.py) and Sorscher et al. 2019 (https://github.com/ganguli-lab/grid-pattern-formation/blob/master/scores.py). We also added a more detailed explanation to our Appendix.
>
> > Pie charts (figure 1) are an inefficient, outdated, and bias-prone visualization method. An empirical CDF or histogram of grid scores or a scatter plot of position error vs. grid score would convey much more information to the readers.
>
> Thank you for this suggestion. We have replaced the pie charts with several more kernel density estimate plots of grid scores to the main paper. If you would like, we can add a scatter plot of position error vs. grid score to our Appendix, although we did not find it particularly illustrative.

---

> > ### Comment · Reviewer_5qo3 · 2022-08-08
> > **Response to authors response**
> >
> > I thank the authors for their detailed response. I went over the updated paper and believe it now deserves a "strong accept". I still think that the perspective taken is somewhat overly pessimistic, but we should not gatekeep criticism.
> >
> > Regarding the code repo: I will not review the code before the NeurIPS decision, but I trust the authors to include the link in the final version to facilitate post-publication peer review.

---

> ### Author Response · Authors · 2022-08-02
> **Response to 5qo3 (1/2)**
>
> Thank you for your detailed review! We are delighted you found our paper to be a “systematic and well presented” “deep, quantitative critique of current connectionist approaches to explaining grid cells.”
>
> In response to your thoughtful criticisms and those of the other reviewers, we have rewritten significant portions of our paper, so we ask that you please take a fresh look. In particular, we rewrote or revised the Discussion, Results (section on match to neural data), Introduction, and (to a lesser extent) Abstract, revamped many of our figures to improve clarity and added many additional sections to our Appendix. We also increased the number of trained networks from ~500 to ~5000.
>
> In response to your specific questions and concerns:
>
> > I am left unconvinced by the claim of many-to-one mappings of training tasks and activity patterns (figure 8 top); the empirical results seem to demonstrate the opposite pattern - surprisingly high sensitivity of grid-cell emergence to the specific operationalization of the training objective
>
> We thank the reviewer for this comment. In fact, both parts of the figure 8 hold, and we should explain this a bit better. The high sensitivity of grid-cell emergence on specific parameter choices is an example of the lower part of Figure 8: a single model, under small perturbations, can yield different solutions. The upper part of Figure 8 can be seen to hold from the different ANN models that can produce grid cells: in Banino et al., the parameters and architecture are different from those in Sorcher et al. and Nayebi et al., and Cueva & Wei. For instance, very high dropout together with gaussian readouts produce grid cells in Banino. By contrast, DoS readouts without dropout produce grid cells in Sorcher et al. These models are examples of how different models may share a common minimum (one where grid cells arise as a solution).
>
> > Therefore, I believe it would be fairer and more constructive to frame the paper's results as a description of a considerable remaining modeling gap and avoid sweeping statements such as "...that one often gets neither"
>
> We agree with this suggested rephrasing and rewrote our Abstract, Introduction, Results and Discussion sections accordingly.
>
> > Why shouldn't we think about the specificity of grid-cell-like representation emergence as a feature rather than a bug of deep neural network modeling? (see above).
>
> This is a great question! We added our answer to our Results & Discussion sections. As you know, previous works claimed that the task of path integration creates grid cells, whereas our paper shows that very specific properties of networks’ supervised targets (single-field single-scale uniformly-distributed place cells with isotropic difference-of-Softmax tuning curves) are necessary to produce grid-like representations. However, these properties are the antithesis of biological place cells; biological place cells are not single field, not single scale, not uniformly distributed, not isotropic, and do not display difference-of-Softmax tuning curves. In fact, experimentally measured place cells are at the opposite extreme e.g. they are heterogeneous with multiple scales in several species (Eliav et al. Science 2021 for bats, Rich et al. Science 2014 for rodents), they do not uniformly cover space, tending to cluster around boundaries and landmarks. We also show in our paper that the path integrating networks learn grid units that are inconsistent with biological grid cells (e.g., networks learn only a single grid module). For these reasons, we believe that the specificity of the emergence of grid-cell-like representation is more likely to be a bug than a feature.
>
> > Is there a repository with the code required for reproducing the study's experiments and results? Adhering to open research practices would increase the impact of this work (and my assigned NeurIPS score). It would also be fairer, given that this study was enabled by the open research practices of the criticized works.
>
> We agree with you 100% regarding open research practices, and we will publicly share our repository and W&B sweeps once the review process concludes. We added this text to our manuscript and inserted a link placeholder (end of Introduction).
>
> We have not yet made our code public because our repository is filled with deanonymizing information (e.g., GitHub usernames, commit messages, comments & TODOs, the README, SLURM script partitions), and NeurIPS treats deanonymization with an immediate rejection. If you would like to review our code during the review process, we could scrub our repository of references to our identities and post a Google Drive link to the zipped code; would you like us to do this?

---

### Public Comment · ~Surya_Ganguli1 · 2022-11-18
**A critical assessment of the claims made in this paper**

We made a critical assessment of the claims made in this paper (the version on OpenReview as of Nov 18th 2022) which we will henceforth refer to as reference [1].

We wrote up our assessment in detail in reference [2]: https://www.biorxiv.org/content/10.1101/2022.11.14.516537v1 by Ben Sorscher, Gabriel Mel, Aran Nayebi, Lisa Giocomo, Dan Yamins, and S.G.

In our assessment we find that the results in [1], when correct, are entirely consistent with prior theoretical work in reference [3]: https://www.sciencedirect.com/science/article/pii/S0896627322009072
and in reference [4]:
https://proceedings.neurips.cc/paper/2019/hash/6e7d5d259be7bf56ed79029c4e621f44-Abstract.html

We further find several errors in [1].  We summarize our results below (detailed explanations in [2]):

(1) In [1] the fact that training ~11,000 path integrators results in hexagonal grid cells only 10% of the time is used to suggest that the emergence of grid cells in trained path-integrators is a more fragile phenomenon than previously reported.  However, although not explained in [1], this result is entirely predicted by theory in [3,4] based on choices made in [1].  In particular, it was already known in [3,4] that 2 out of the 4 nonlinearities used in [1] should lead to hexagonal grid cells.  Also it was already known in [3,4] that only 1/5 place cell structures used in [1] should lead to hexagonal grid cells.  The fact that 2/4 * 1/5 = 1/10 thus explains why [1] only found hexagonal grid cells 10% of the time given the choices they made.  Moreover when [1] makes two key choices shown to be important in [3,4], they robustly obtain hexagonal grids close to 100% of the time.  Thus when the results of [1] are evaluated within the proper context of prior theoretical work, the results are completely expected:  hexagonal grid cells robustly emerge, without any fine-tuning, precisely when prior theory predicts they should, and don’t when prior theory predicts they should not.

(2) [1] claims that difference of Gaussian (DoG) place cells cannot yield hexagonal grid cells. This is incorrect. See Sec. 4 of [2].

(3) [1] suggests that place cells with multiple firing fields cannot lead to hexagonal grid cells.  We demonstrate that they can as long as they are dominated by a single scale (See Sec. 5 of [2]), as predicted by theory in [3,4].  The motivation for studying one scale at time comes from the topographic organization of scale along the dorsoventral axis.

(4) [1] claims that the emergence of hexagonal grid cells is extremely sensitive to small changes in place cell scale. Such sensitivity is inconsistent with prior theory in [3,4].  We thus performed our own robustness check and we do not see this sensitivity (compare Fig. 4B with Fig. 4A in Sec. 6 of [2]).  Our interpretation is that the sensitivity seen in [1] may simply reflect statistical noise in grid scores in path-integrators that do not robustly exhibit hexagonal grid cells to begin with (see Sec 6 of [2] for details).

(5) [1] presents a proof that Gaussian place cells cannot generate periodically patterned cells in trained path integrators. We show they can in Sec. 7 of [2]. We resolve the discrepancy between the claimed proof in [1] and our simulations by showing how the proof in [1] is incomplete and how the theory in [3, 4] suggests periodic patterns are indeed possible with Gaussian place cells.

(6) [1] claims that DoG/DoS structure in place cells is biologically unrealistic. We discuss in Sec. 8 of [2] that when adhering to the precise interpretation of DoG/DoS structure, given in [4], as that of grid cell inputs to place cells, rather than the rectified outputs of place cells, such DoG/DoS structure is not obviously implausible given the current lack of the requisite experiments to rule it out.

---

### Public Comment · ~Rylan_Schaeffer2 · 2022-11-20
**Clarification Regarding Response to Reviewer, Concerning Neural Data Availability**

In a response to a reviewer, regarding Nayebi et al. 2021, we wrote: "However, the analysis code is not open source and the neural data is not public; we have asked on several occasions for access to the neural data, but we did not receive it." This statement is inaccurate: one of us asked the wrong individuals, who did not have deciding authority for neural data release, for access to the neural data, and were told it could not be made currently available. The PI responsible for the data and their lab have gone to considerable lengths to make their data publicly available, and they would have shared it had we asked directly. We sincerely apologize for our error.

---

### Meta-Review · Area_Chair_6Srq · 2022-08-23

**Recommendation:** Accept
**Confidence:** Certain

**Metareview:**

This paper examines previous deep learning models of the entorhinal-hippocampal formation that have been reported to exhibit the emergence of grid cells. The authors show that this phenomenon only occurs under very specific hyperparameter settings and with a variety of post-hoc modelling decisions, suggesting that it is not a robust finding based on the cost functions used, as suggested in the original work. The authors use this as a case study to make the point that inductive biases and consideration of biological knowledge should be considered critical ingredients for models in computational neuroscience, and that more care needs to be taken when reporting emergent phenomena in deep learning models for neuroscience.

The reviewers were generally positive about the paper, and agreed it makes a worthwhile contribution. They had some concerns related to potential over-reach in the paper's statements/claims, but after engaging with the authors, the majority of such concerns were addressed. As such, consensus was reached to accept this paper.

**Award:**

No

---

### Decision · Program_Chairs · 2022-09-14

Accept